# Ubp2 modulates DJ-1-mediated redox-dependent mitochondrial dynamics in *Saccharomyces cerevisiae*

Sananda Biswas, Patrick D'Silva 🔟 *

Department of Biochemistry, Division of Biological Sciences, Indian Institute of Science (IISc), Bengaluru, Karnataka, India

* patrick@iisc.ac.in

## Abstract

Mitochondrial integrity is a crucial determinant of overall cellular health. Mitochondrial dysfunction and impediments in regulating organellar homeostasis contribute majorly to the pathophysiological manifestation of several neurological disorders. Mutations in human DJ-1 (*PARK7*) have been implicated in the deregulation of mitochondrial homeostasis, a critical cellular etiology observed in Parkinson's disease progression. DJ-1 is a multifunctional protein belonging to the DJ-1/ThiJ/PfpI superfamily, conserved across the phylogeny. Although the pathophysiological significance of DJ-1 has been well-established, the underlying molecular mechanism(s) by which DJ-1 paralogs modulate mitochondrial maintenance and other cellular processes remains elusive. Using *Saccharomyces cerevisiae* as the model organism, we unravel the intricate mechanism by which yeast DJ-1 paralogs (collectively called Hsp31 paralogs) modulate mitochondrial homeostasis. Our study establishes a genetic synthetic interaction between Ubp2, a cysteine-dependent deubiquitinase, and DJ-1 paralogs. In the absence of DJ-1 paralogs, mitochondria adapt to a highly tubular network due to enhanced expression of Fzo1. Intriguingly, the loss of Ubp2 restores the mitochondrial integrity in the DJ-1 deletion background by modulating the ubiquitination status of Fzo1. Besides, the loss of Ubp2 in the absence of DJ-1 restores mitochondrial respiration and functionality by regulating the mitophagic flux. Further, Ubp2 deletion makes cells resistant to oxidative stress without DJ-1 paralogs. For the first time, our study deciphers functional crosstalk between Ubp2 and DJ-1 in regulating mitochondrial homeostasis and cellular health.

## Author summary

Mitochondria are dynamic organelles essential for generating the energy required to maintain cellular viability and drive biological processes. Mitochondrial structures undergo continuous remodeling, modulating their function in response

**Data availability statement:** All data are in the manuscript and/or supporting information files.

**Funding:** This work was supported by DST-SERB (Grant File No. CRG/2018/001988) and Department of Biotechnology (DBT-IISC Partnership Program Phase-II, No. BT/PR27952/IN/22/21/2018) and DST-FIST Programme Phase III (No. SR/FST/LSII-045/2016(G) (to P.D.S). The funders had no role in study design, data collection and analysis, decision to publish, or preparation of the manuscript.

**Competing interests:** The authors have declared that no competing interests exist.

to cellular cues. The plasticity of mitochondrial structures is due to conserved fusion-fission proteins, thus enabling cells to adapt to metabolic changes. Mutations in *PARK7*, encoding for DJ-1, lead to an imbalance in mitochondrial dynamics and culminate in the progression of neurodegenerative disorders such as Parkinson's disease (PD). DJ-1 belongs to the highly conserved *DJ-1*/ThiJ/Pfp superfamily of multifunctional proteins. *Saccharomyces cerevisiae* encodes for four paralogs, which belong to the DJ-1 superfamily. Recent studies demonstrate the role of yeast DJ-1 members in regulating mitochondrial integrity and oxidative stress response. However, the mechanism(s) by which the paralogs mediate cytoprotective action remains elusive. The current study addresses the mechanistic lacuna by delineating cross-talk between Ubp2, a deubiquitinase, and redox-sensitive DJ-1 paralogs in regulating mitochondrial health. Our results suggest that elevated expression of Ubp2 in cells lacking DJ-1 paralogs promotes hyperfused mitochondrial structures. At the same time, in the absence of DJ-1 paralogs, the levels of Fzo1 expression are enhanced significantly due to its altered ubiquitination status. Intriguingly, mitochondrial dynamics and cellular health were reinstated upon deletion of Ubp2, particularly in cells with combinatorial deletion of DJ-1 paralogs in yeast. The study thus provides evidence linking the role of DJ-1 and deubiquitinase in the maintenance of mitochondrial dynamics, which can further aid in understanding the mechanism causing PD progression.

## 1 Introduction

The DJ-1 superfamily comprises multifunctional proteins conserved across the phylogeny [1–3]. They represent a class of proteins essential for the upkeep of cellular metabolism across various species, from prokaryotes to eukaryotes [4–10]. DJ-1 paralogs exhibit acid stress resistance in bacteria and function as glyoxalase and protease [7,11,12]. *Saccharomyces cerevisiae* has four paralogs, namely, Hsp31, Hsp32, Hsp33, and Hsp34, collectively called the Hsp31 paralogs, which belong to the DJ-1 superfamily. These paralogs are involved in metabolic reprogramming and cellular survival in the stationary phase [8,13]. Moreover, the Hsp31 paralogs regulate the redox status of the cells and are essential for combatting carbonyl and concomitant oxidative stress [14–16]. Under oxidative stress conditions, the Hsp31 paralogs redistribute to mitochondria and provide cytoprotection by their deglycase, glyoxalase, and chaperoning activities [5,14,16]. Additionally, reports suggest that Hsp31 paralogs remodel the mitochondrial pool and modulate the etiology of protein aggregation in models of neurodegenerative disorders such as Parkinson's disease (PD), Alzheimer's disease (AD), and Huntington's disease [5,13,15].

In humans, mutations in *PARK7* (encodes human DJ-1) are responsible for the progression of familial forms of PD [17–20]. Comparable to its orthologs from other genera, several biochemical studies also demonstrate the neuroprotective functions

of human DJ-1 due to its functions as glyoxalase, protease, and a chaperone [10,21–23]. Moreover, overexpression of the pathogenic mutants of DJ-1 leads to mitochondrial fragmentation and dysfunction and enhanced sensitivity to oxidative stressors [24]. Notably, it is known that DJ-1 homologs in yeast, humans, and plants redistribute to mitochondria in response to oxidative stress to safeguard organellar function [9,14,16]. Furthermore, overexpressing human DJ-1 in yeast reduced mitochondrial superoxide levels and functionally supplemented Hsp31, demonstrating the functional conservation between species [1,14]. However, even though DJ-1 has been identified as a crucial modulator of mitochondrial integrity across the genera, its mechanism of regulating organellar health remains elusive.

Mitochondria are dynamic organelles that undergo cycles of fusion and fission in response to various metabolic cues in the cell. For instance, the process of mitochondrial fission is necessary for maintenance of mitochondrial quality control, whereas mitochondrial fusion facilitates mixing of intramitochondrial contents necessary for maintaining mitochondrial functions [ 25]. The PD-associated DJ-1 mutants cause an imbalance in mitochondrial dynamics by affecting the expression of Drp1, a conserved GTPase required for mitochondrial fission [26]. Similarly, in *Saccharomyces cerevisiae*, cells lacking Hsp31 paralogs cause perturbed mitochondrial dynamics due to altered expression of Fzo1 [15]. Fzo1 is the critical mediator of mitochondrial fusion and is well-established among the dynamin family proteins [27]. An orchestrated action of inner (Mgm1) and outer membrane proteins (Fzo1, Ugo1) regulates fusion in yeast [28]. These evolutionarily conserved GTPase activities are regulated by several post-translational modifications, including proteolytic processing, ubiquitylation, sumoylation, phosphorylation, and dephosphorylation [29]. Tethering of neighboring mitochondrial membranes is initiated by the oligomerization of Fzo1, eventually leading to the fusion of the outer membrane [30]. Ubiquitination is one of the critical processes that regulate the fusogenic activity of Fzo1 on the mitochondrial outer membrane [31]. Their ubiquitination status dictates the fate of mitochondrial fusion proteins in the outer membrane. The ubiquitination-deubiquitination pathway determines the homeostatic balance of fusion and fission [32]. It is well-known that ubiquitination directs proteins to the proteasomal complex for degradation, and recent studies indicate that there can be many other biological outcomes owing to the diversity of ubiquitin chain linkages [33,34]. Deubiquitinases (DUBs) constitute a class of enzymes that catalyze the process of removing ubiquitin from ubiquitinated proteins, thus rescinding the ubiquitination process [35]. The contrasting action of deubiquitinases is a crucial regulatory step in this complex process [32]. Fzo1 turnover is regulated by a class of cysteine-dependent proteases with opposing actions. Ubp2 and Ubp12 are the thiol-dependent, ubiquitin-specific proteases with antagonistic functions in stabilizing Fzo1 [32]. Notably, the ubiquitination status of mitofusins is a significant determinant of mitochondrial dynamics, which in turn helps cells adapt to a variety of conditions [36,37]. Ubiquitin chain editing by the deubiquitinases adds another level of regulation that dictates the status of mitofusins. The action of Ubp2 is essential for stabilizing the 'pro-fusion' ubiquitin chain on Fzo1. Thus, cells lacking Ubp2 display fragmented mitochondrial morphology. However, another thiol-dependent deubiquitinase, Ubp12, removes the 'pro-fusion' ubiquitin tags from Fzo1 and directs it to proteasomal degradation [38]. The balance of these antagonistic DUBs is maintained by a cell division cycle protein, Cdc48 [32]. This mounting evidence thus suggests a plausible cross-talk between Hsp31 paralogs and the deubiquitinase cascade by which mitochondrial dynamics are regulated.

By utilizing *Saccharomyces cerevisiae* as the model system, the current study outlines the mechanism by which Ubp2 and the redox-sensitive DJ-1 paralogs functionally cooperate to maintain mitochondrial health and integrity. Ubp2, a deubiquitinase required for retaining the fusogenic population of Fzo1, was identified to interact genetically with yeast DJ-1 members, particularly Hsp31 and Hsp34. The mitochondrial structure and function were restored upon deletion of Ubp2 in the combinatorial absence of Hsp31 and Hsp34. Moreover, the deletion of Hsp31 paralogs sensitizes the cells to oxidative stress. However, the oxidative stress sensitivity was partially alleviated in cells lacking Ubp2 in conjunction with Hsp31 paralogs. Taken together, our findings provide evidence for delineating a functional cross-talk between Hsp31 paralogs and Ubp2 by altering the ubiquitination status of Fzo1, which regulates mitochondrial dynamics.

## 2 Results

### 2.1 Deletion of Ubp2 restores respiratory growth defects in the absence of Hsp31 paralogs

Mitochondrial structure and function are essential indices of cellular health and have been studied extensively owing to the dynamicity of the organelle. The Hsp31 paralogs in *Saccharomyces cerevisiae*, which belong to the DJ-1 superfamily, have been identified as crucial modulators of mitochondrial health [14,15]. It has been reported that a combinatorial deletion of Hsp31 and Hsp34 results in a growth defect in glycerol at non-permissive temperatures, suggesting the role of Hsp31 paralogs in maintaining mitochondrial integrity [15]. Alteration in the expression of Fzo1 due to the absence of Hsp31 paralogs manifests as an increase in hyperfused mitochondrial structures. However, the mechanism by which these paralogs perturb the mitochondrial dynamics remains elusive.

The process of fusion is a finely orchestrated mechanism mediated by the action of a conserved class of cysteine-dependent deubiquitinases. Recent evidence suggests that Ubp2 is a crucial modulator of mitochondrial dynamics and essential for maintaining the fusogenic status of Fzo1 [38]. Since Hsp31 paralogs regulate Fzo1 expression and mitochondrial dynamics, we investigated a possible functional cross-talk between Ubp2 and Hsp31 paralogs. Using yeast genetic manipulation, we created Δ*ubp2* in the BY4741 strain background. The deletion of Ubp2 (Δ*ubp2*) leads to an impairment in the growth in both complete (S.C. Dextrose) and non-fermentable media (S.C. Glycerol) at non-permissive temperature of 37°C (Fig 1A). At the same time, among the four Hsp31 paralogs, only the cells lacking Hsp31(Δ*31*) exhibit a slight reduction in growth in glycerol at a non-permissive temperature of 37 °C (Fig 1A) [15]. The cells lacking Hsp34 (Δ*34*) have growth comparable to WT cells. However, together deletion with Hsp31 (Δ*31Δ34*) leads to severely compromised growth in glycerol in conditions of heat stress at 37°C (Fig 1A). To test a genetic interaction between Hsp31 paralogs (Hsp31 and Hsp34) and Ubp2, we constructed combination of deletion strains lacking Ubp2 in the background of Δ*31*(Δ*31Δubp2*), Δ*34*(Δ*34Δubp2*) and Δ*31Δ34* (Δ*31Δ34Δubp2*). Interestingly, the strains Δ*31Δubp2* and Δ*34Δubp2* alone had a more compromised growth phenotype in glycerol at 37°C, comparable to Δ*ubp2* (Fig 1A). Intriguingly, on the other hand, the deletion of Ubp2 in the absence of 31 paralogs (Δ*31Δ34Δubp2)*showed rescue in the growth phenotype comparable to WT in a non-fermentable carbon source and non-permissive temperature of 37°C (Fig 1A).

To determine whether the growth rescue by the Ubp2 in the absence of Hsp31 paralogs is due to the consequence of its regulated expression, we first analyzed the levels of Ubp2 in WT and Δ*31Δ34* strains. To test this, Ubp2 was genomically tagged with hemagglutinin (HA) at the C-terminus and expressed in WT and Δ*31Δ34*. The whole cell lysates were subjected to immunoblotting and probed for the Ubp2 levels. The immunoblot analysis revealed that Δ*31Δ34* strains showed up to a 2.5-fold increment in the expression of Ubp2 levels in cells from late log phase (Fig 1B and 1C). While, WT cells exhibit a significant reduction in Ubp2 expression in the late log phase compared to the cells collected from the early-log phase (S1B Fig, compare lanes 1 and 3),Ubp2 expression remained unaltered in Δ*31Δ34* across mid-log (6h) and late-log (12h) phases (S1B Fig, compare lanes 2 and 4), suggesting Hsp31-dependent regulated expression of Ubp2. To further ascertain this observation, the Ubp2 as the molecular player in regulating the growth defect in Δ*31Δ34*, we overexpressed Ubp2 (under TEF promoter) in the background of Δ*ubp2*, Δ*31Δ34Δubp2*, and subjected the strains to growth analysis. The overexpression of Ubp2-HA was ascertained by immunoblotting (S2 Fig). The phenotypic assessment in glycerol indicated that Ubp2 overexpression in Δ*ubp2* shows a phenotype comparable to WT. In contrast, Ubp2 overexpression in Δ*31Δ34 leads* to growth sensitivity in glycerol at 37°C, equivalent to Δ*31Δ34* alone (Figs 1D and S16). These experimental evidence suggest a strong functional cross-talk between Hsp31 paralogs and Ubp2 in terms of regulating respiratory growth in conditions of heat stress.

### 2.2 Ubp2 regulates mitochondrial integrity and turnover in cells lacking Hsp31 paralogs

The phenotypic growth defects in respiratory deficient conditions are closely correlated to perturbance of mitochondrial health. Therefore, we assessed mitochondrial-specific functional parameters across all the strains. In particular, the

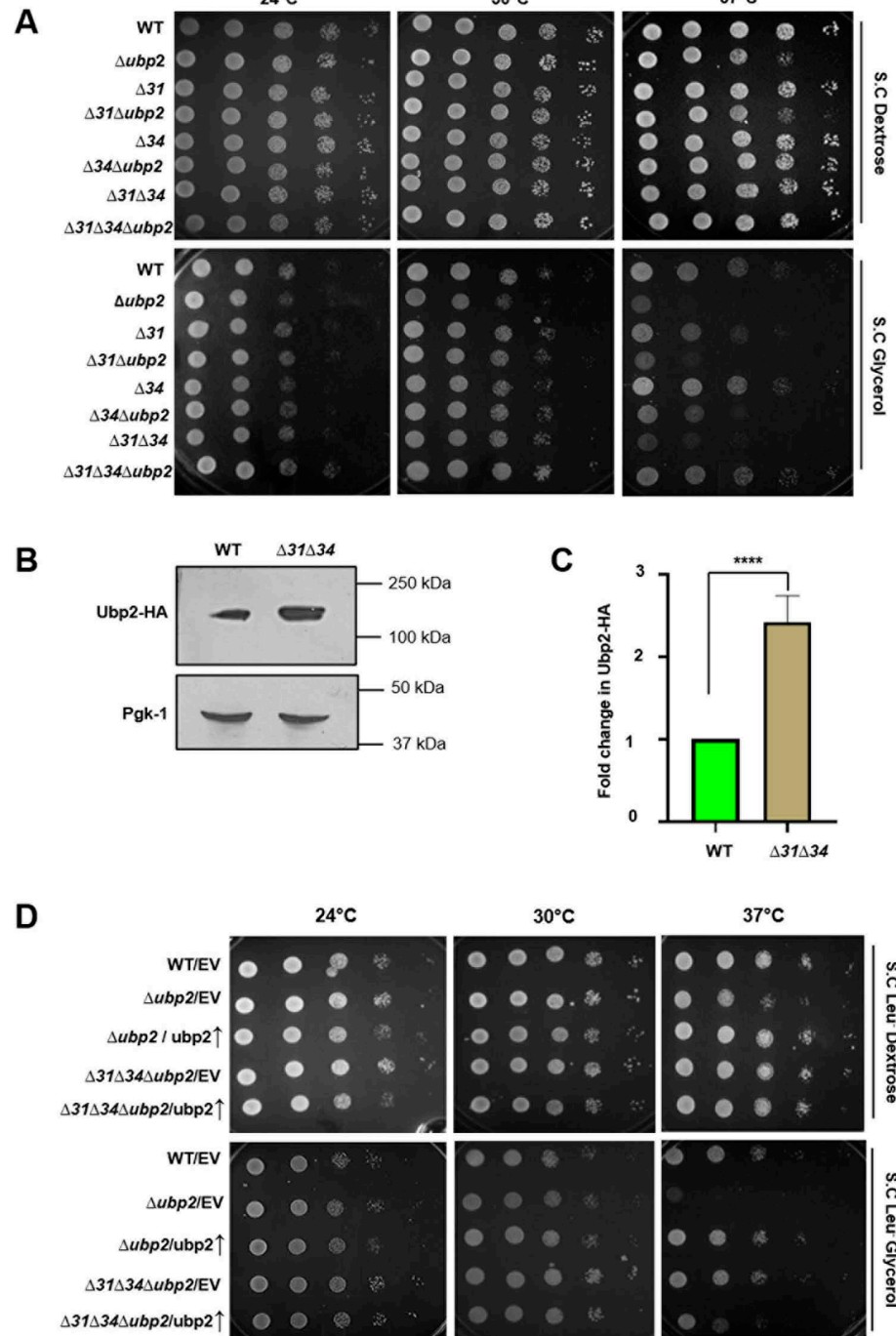

**Fig 1. Ubp2 deletion restores the respiratory growth defect of Δ31Δ34 strain. (A)** Growth phenotype assessment. The indicated yeast strains were allowed to grow up to mid-log phase in SC dextrose broth at 30°C. Ten-fold serially diluted cells were spotted on the indicated media and incubated at permissive temperature (30°C) and non permissive temperature (24°C and 37°C). Images were captured at 36h and 72h for dextrose and glycerol, respectively. **(B)** Assessment of the Ubp2-HA levels in the absence of Hsp31 paralogs. Whole-cell lysate of the indicated strains was subjected to immunoblotting and probed for Ubp2-HA levels in late log phase(12h). **(C)** Densitometric analysis for probing the difference in expression of Ubp2-HA levels. **(D)** Growth phenotype assessment upon complementation with Ubp2. The indicated strains were allowed to grow up to the mid-log phase in SC Leu- media and subjected to grow at permissive and non-permissive temperatures in dextrose and glycerol. Images were captured for dextrose and glycerol after 36h and 72h, respectively. Unpaired Student t-test was performed for statistical analysis. Error bars represent the standard deviation in median values from 3 biological replicates. Asterisks indicate the p-value, *, $p < 0.05$; **, $p < 0.01$; ***, $p < 0.001$; ****, $p < 0.0001$.

changes in the mitochondrial morphology are assessed to ascertain the overall cellular health. To microscopically visualize mitochondrial structures, the cells were transformed with MTS-mCherry constructs that specifically translocate to the organelle due to targeting sequence and decorate the mitochondria [14]. In agreement with the previous findings, the Δubp2 strain exhibited fragmented mitochondria [39] (Fig 2A, *compare panels 2 with panel 1*). In contrast, the Δ31Δ34 strain showed hyperfused mitochondrial morphology [15] (Fig 2A, *compare panels 3 with 1*). Most interestingly, upon deletion of Ubp2 in Δ31Δ34 background, mitochondrial morphology was reversed, which now closely resembles the WT cells (Fig 2A, *compare panels 4 with panel 1*). The fragmented, intermediate, and tubular states of the mitochondria were subjected to quantification for each respective strain. The results highlight that the cells exhibiting intermediate mitochondrial morphology in Δ31Δ34Δubp2 are comparable to that of the WT in correlation with the reversal of growth phenotype (Fig 2B) compared to the extensive tubular network observed in Δ31Δ34 strain. On the other hand, Ubp2 deletion in the background of single deletion strains of Hsp31 paralogs (Δ31Δubp2, Δ34Δubp2) exhibited fragmented mitochondria, comparable to the Δubp2 strain (S3 Fig). Further, to validate the genetic interaction between Ubp2 and Hsp31 paralogs, Ubp2 was overexpressed in Δ31Δ34Δubp2, and the mito*chondrial* network was assessed.Corroborating with the phenotype, Ubp2 overexpression in Δ31Δ34Δubp2 restores the hyperfused mitochondrial structures *c*ompared to WT (S12 Fig, *compare panels 1 and 5*). Intriguingly, *the* catalytically inactive form of Ubp2 (ubp2$_{C745S}$) could not restore the hyperfused network of mitochondria (S12 Fig, *compare panels 1,5 and 6*), highlighting the role of catalytic cysteine in the maintenance of mitochondrial dynamics.

As we observed varied mitochondrial structures across the different strains, we probed for the total and functional mitochondrial mass change using flow cytometric analysis. NAO (10-N-nonyl-acridine orange) is used specifically for staining cardiolipin, an indicator of the total mitochondrial mass in cells. Upon flow cytometry, deletion of Ubp2 did not show a significant difference in the total mitochondrial mass compared to WT (Fig 2C). However, the Δ31Δ34 strain exhibited a substantial increase in the total mitochondrial mass, as reported previously [15]. Interestingly, there was no difference in the total mitochondrial mass in Δ31Δ34Δubp2, and it was comparable to Δ31Δ34 (Fig 2B). This indicates that deletion of Ubp2 in the background of WT and Δ31Δ34 does not alter the total mitochondrial mass. The functionality of mitochondria was assessed by staining the cells with a potentiometric dye, JC-1, and subjecting them to flow cytometric analysis. A significant decrease in the functional mitochondrial content in Δubp2 *was* observed compared to WT (Fig 2D). Contrastingly, the Δ31Δ34 strain exhibited a significant increase in the functional mitochondrial mass [15], as reported previously. Strikingly, the deletion of Ubp2 in the background of Δ31Δ34 restored the functional mitochondrial mass comparable to WT (Fig 2D). In order to validate the result, Ubp2 was overexpressed in the background of Δ31Δ34Δubp2 and the total and functional mitochondrial mass was measured (S13A and S13B Fig). The total mitochondrial mass upon overexpression of Ubp2 in Δ31Δ34Δubp2 is comparable to Δ31Δ34, consistent with the previous results(S13A Fig). Moreover, the functional mitochondrial mass increases upon Ubp2 overexpression in Δ31Δ34Δubp2 and is comparable to Δ31Δ34 (S13B Fig).

Furthermore, as mitochondrial function is closely associated with respiration status, leading to ATP generation, we measured the mitochondrial ATP levels across all the strains. Corroborating the previous data, we found a significant decrease in ATP levels in cells lacking Ubp2, while Δ31Δ34 *exhibit* higher ATP levels. At the same time, the ATP levels in Δ31Δ34Δubp2 have restored to WT, consistent with the reversal of mitochondrial-specific phenotypes correlating to the suppression of growth defects (Fig 2E). These results suggest that Ubp2 plays a crucial role in maintaining functional mitochondrial mass in conjunction with Hsp31 paralogs.

Mitochondrial homeostasis is regulated by mitophagy, which clears the damaged and aged mitochondria [40]. We observed that loss of Ubp2 does not affect the total mitochondrial mass, albeit involving the functional mitochondrial pool. We assayed the strains for organellar turnover to investigate whether Ubp2 has a role in modulating mitophagy. Mitophagy is monitored by tagging the outer mitochondrial membrane protein (OM45) with GFP at the C-terminus and subjecting the strains to microscopic and immunoblotting analysis [41]. The cells were grown in dextrose until the mid-log phase and then in glycerol for the subsequent time points as indicated. Samples were collected, and cells

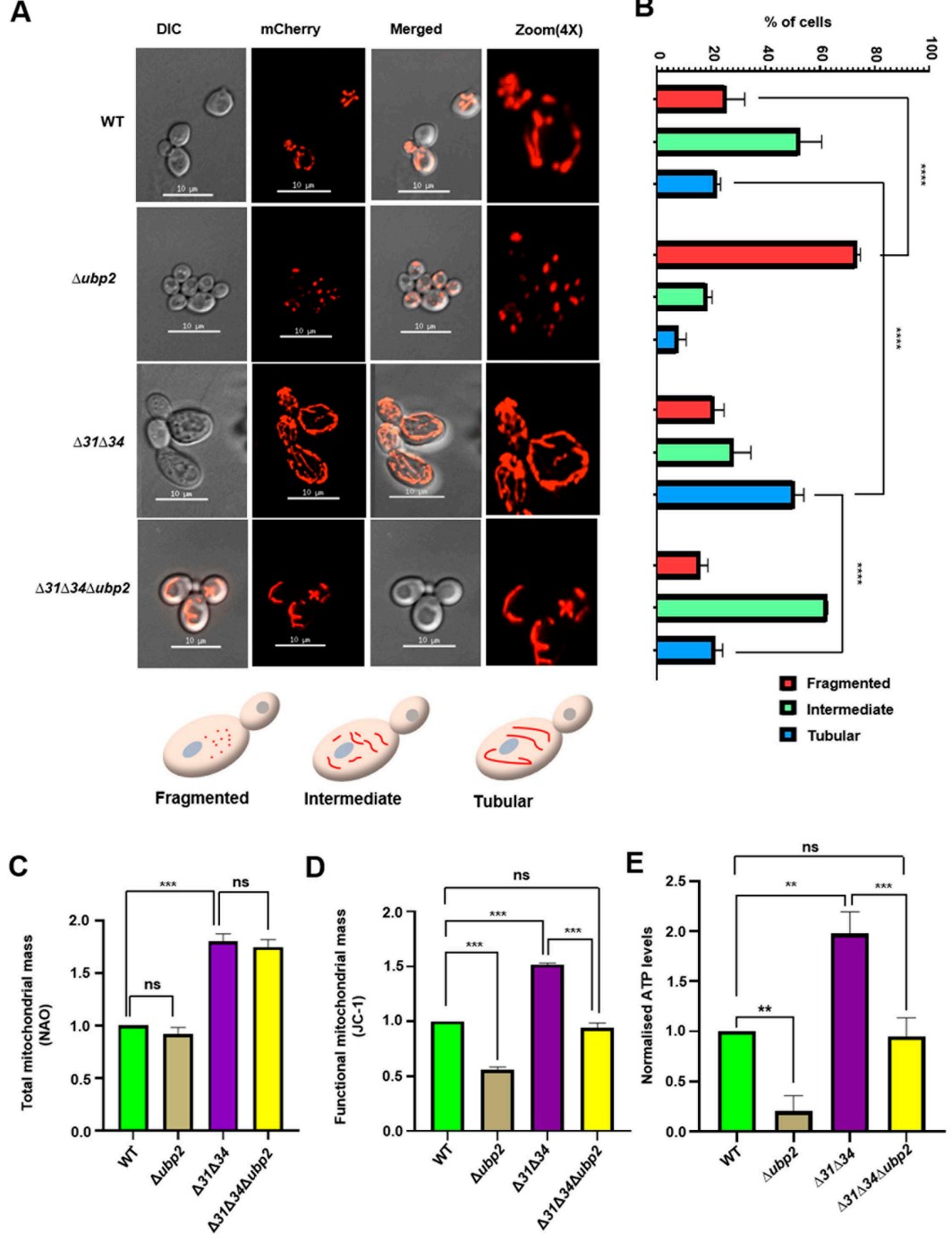

**Fig 2. Ubp2 deletion reverts the defect in mitochondrial integrity in the absence of Hsp31 paralogs. (A)** Assessment of mitochondrial integrity. Yeast strains expressing MTS-mCherry grown in SC Leu⁻ dextrose till the mid-log phase were subjected to microscopic analysis to visualize mitochondria. Scale bar (10 μm). The three different mitochondrial morphologies scored have been represented (Fragmented, Intermediate, Tubular cells). **(B)** Quantification of cells exhibiting specific mitochondrial morphology. **(C)** Quantification of total mitochondrial mass by flow cytometric analysis. Cells grown until the mid-log phase were subjected to NAO staining and quantified using the BD FACS Verse instrument. **(D)** Evaluation of the functional mitochondrial mass by flow cytometry. Cells grown until the mid-log phase were subjected to JC-1 staining, a potentiometric dye, and acquired using BD FACS Verse. **(E)** Measurement of ATP levels. Mitochondria were isolated from the indicated strains, and ATP was measured using a fluorescence-based assay. One-way ANOVA with Tukey's multiple comparison test was used for significance analysis. Error bars represent the standard deviation in median values from 3 biological replicates. Asterisks indicate the p-value, *, $p < 0.05$; **, $p < 0.01$; ***, $p < 0.001$; ****, $p < 0.0001$.

were lysed for analysis using immunoblotting. The results suggest that the process of mitophagy is initiated in the WT strain after 36h of mitophagy induction (Fig 3A). On the other hand, the Δubp2 exhibits mitophagy initiation at 24h of induction, probably due to the presence of fragmented smaller mitochondrial structures. However, the intensity of the processed GFP in Δubp2 is less than the WT at the same time points. This indicates that even though mitophagy is initiated at an earlier time in Δubp2, clearance of damaged mitochondria is slower than the WT, accumulating fragmented and nonfunctional mitochondrial mass. Δ31Δ34 exhibits comparable mitophagy rate to WT, albeit at a lower processing efficiency. However, Δ31Δ34 showed slower mitophagy induction possibly due to the presence of hyperfused mitochondrial structures (Fig 3A and 3B). Interestingly, Δ31Δ34Δubp2, which has restored mitochondrial integrity exhibits kinetically faster mitochondrial clearance by induction observed at 36h as compared to Δ31Δ34 alone (Fig 3A and 3B). However, the intensity of processed GFP is less than Δ31Δ34 and not comparable to WT, suggesting that Ubp2 deletion partially suppresses the process of mitophagic flux. The mitophagy deficient Δatg32 strain served as a negative control for GFP processing (Fig 3A, *last panels*).

To validate the results further, we also subjected the strains to microscopy to observe the vacuolar internalization of processed GFP, after staining with FM4–64 to visualise the vacuole. In correlation with the immunoblotting results, Δubp2 exhibits processed GFP comparable to WT. Still, the fluorescence intensity was higher than WT after 48h of mitophagy induction (Fig 3C, *compare panels 1 and 2*). Similarly, OM45-GFP is internalized to a lesser extent in Δ31Δ34 compared to Δ31Δ34Δubp2 (Fig 3C, *compare panels 3 and 4*). However, the level of processed OM-45 GFP fluorescence intensity in Δ31Δ34Δubp2 is lesser than WT, indicating that Ubp2 partially restores the mitophagic turnover. The cells are deficient in mitophagy (Δatg32), did not show any processed OM-45 GFP fluorescence in the vacuole, and were used as a negative control (Fig 3C, *compare panels 1–4 with 5*).

## 2.3 Fzo1 levels and turnover are restored upon deletion of Ubp2 in the absence of Hsp31 paralogs

The expression and stability of proteins like Fzo1 and Dnm1 in *Saccharomyces cerevisiae* determine the dynamicity of the mitochondria [42]. Our results exclusively highlight variations in mitochondrial health, and we analyzed for changes in the expression of mitochondrial dynamic proteins. In agreement with previous findings, we observed a significant decrease in Fzo1 levels in Δubp2 compared to WT (Fig 4A and 4B). In contrast, a substantial increment in the expression of Fzo1 was observed in Δ31Δ34 strains [15] (Fig 4A and 4B). Interestingly, Δ31Δ34Δubp2 exhibited a restoration in levels of Fzo1, comparable to WT (Fig 4A and 4B). To ascertain the role of Ubp2 in stabilizing Fzo1 and promoting fusion, Ubp2 overexpressed in the WT background, and phenotypic analysis was performed. In agreement with the previous results, WT/Ubp2(↑) exhibits a compromised phenotype in glycerol at 37°C (S5A Fig). At the same time, mitochondria exhibited hyperfused morphology upon Ubp2 overexpression (S5B Fig). The hyperfusion of mitochondria results from enhanced expression of Fzo1 (S5C Fig).

To validate the genetic interaction between Ubp2 and Hsp31 paralogs, Ubp2 overexpressed in Δ31Δ34Δubp2, and the steady-state levels of Fzo1 were analyzed by immunoblotting. The results suggest a significant increment in Fzo1 expression upon overexpression of Ubp2 in Δ31Δ34Δubp2 as compared to WT (S6A Fig). Furthermore, there were no significant differences in the expression of fission protein Dnm1 across the strains (S15 Fig). The results further suggest that the absence of both Hsp31 paralogs and Ubp2 keeps the levels of Dnm1 unaltered in the cells, which is in agreement with the previous findings [15,39].

As there was a restoration of Fzo1 steady-state levels in Δ31Δ34Δubp2, we tested for the turnover rate of Fzo1 across the four strains by utilizing the cycloheximide chase assay (CHX). The turnover kinetics revealed that Fzo1 has a shorter half-life in Δubp2 compared to WT (Fig 4C and 4D). As opposed to Δubp2, the Δ31Δ34 strain exhibited stable expression of Fzo1, correlating with the increased steady-state expression (Fig 4C and 4D). However, Δ31Δ34Δubp2 exhibits restoration of Fzo1 dynamics, comparable to WT (Fig 4C and 4D). Furthermore, upon overexpression of Ubp2 in Δ31Δ34Δubp2, the turnover kinetics of Fzo1 was found comparable to WT (S6B Fig).

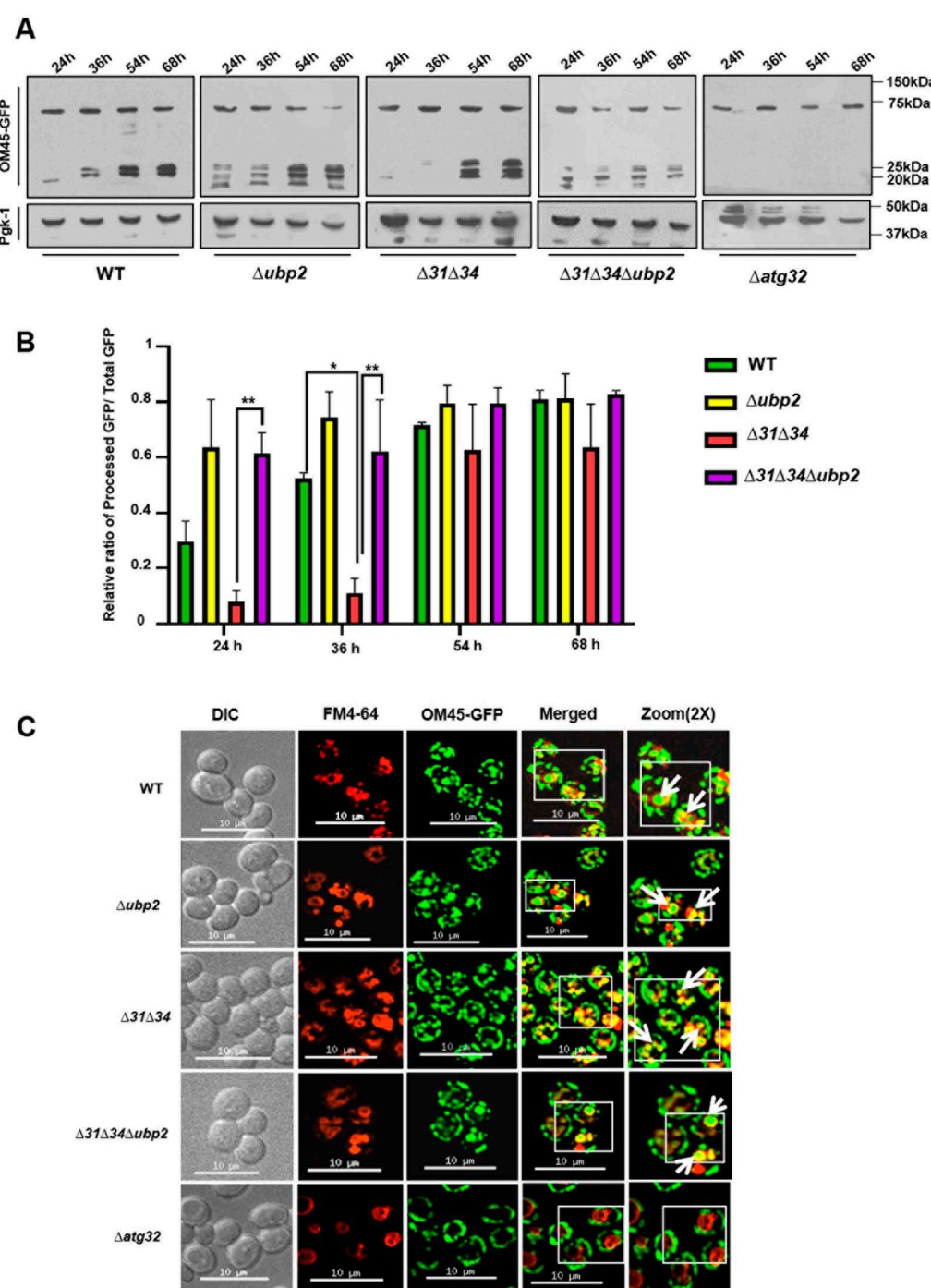

**Fig 3. Role of Ubp2 in regulating mitophagy. (A, B)** Mitophagy induction by Western analysis. Cells expressing OM45-GFP from the indicated strains were cultured in glycerol for the mentioned time points, and the lysates were subjected to Western blotting and quantified using Two-Way ANOVA for significance analysis. Error bars represent the standard deviation in median values from 3 biological replicates. Asterisks indicate the p-value, *, $p < 0.05$; **, $p < 0.01$; ***, $p < 0.001$; ****, $p < 0.0001$ **(B)**. Δatg32 was used as a positive control to compare the defects in mitophagy. **(C)** Microscopic analysis of mitophagy induction. Cells expressing OM45-GFP were grown in YP-glycerol for 48h and subjected to fluorescence microscopy. The vacuole is stained with FM4-64 dye. Scale bars (10 μm). Images were zoomed up to 2X and represented.

PLOS Genetics

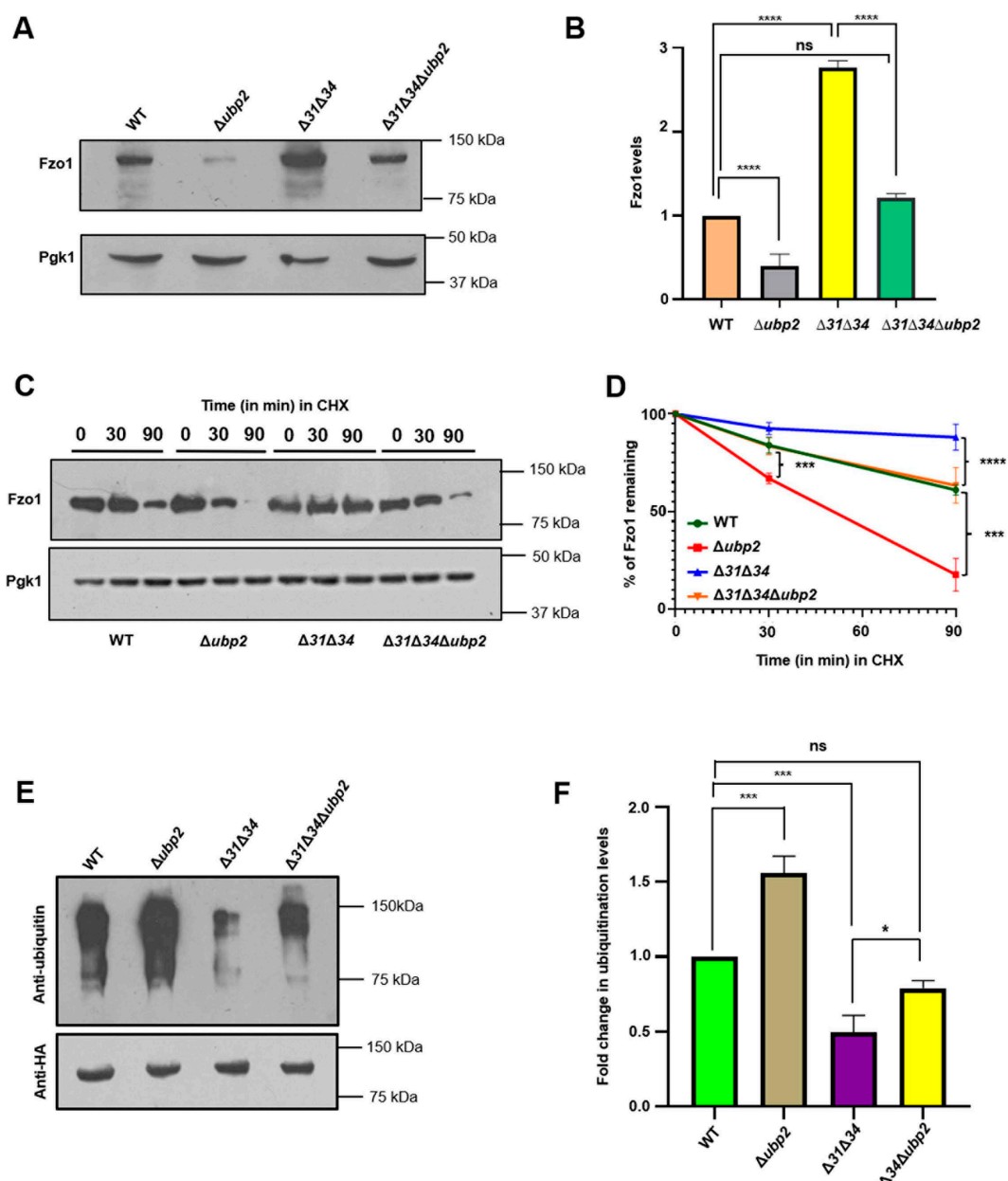

**Fig 4. Fzo1 levels are restored upon deletion of Ubp2 in the absence of Hsp31 paralogs. (A, B)** Evaluation of the relative steady-state levels of Fzo1 by western blotting. The cells from indicated strains lysed, and Fzo1 levels were probed in the whole cell lysate **(A)**. The blots were subjected to quantification by densitometry, and the fold change in Fzo1 expression was represented. **(C, D)** Assessment of the turnover kinetics of Fzo1 using cycloheximide assay. The cells were subjected to treatment with cycloheximide for the indicated time points, and the lysates were subjected to western blotting **(C)**. The blots were subjected to quantification densitometrically and represented the rate at which Fzo1 is degraded across the strains **(D)**. Two-way ANOVA was performed using three biological replicates. **(E, F)** Evaluation of ubiquitination status of Fzo1. Genomically tagged Fzo1 at the C-terminus with HA-tag was subjected to pull-down analysis using HA-conjugated beads and probed with anti-HA and anti-ubiquitin specific antibodies **(E)**. The Western blots were subjected to quantitation by densitometry, and fold change in the ubiquitination levels is represented **(F)**. One-way ANOVA with Tukey's multiple comparison tests was used for significance analysis. Error bars represent the standard deviation in median values from 3 biological replicates. Asterisks indicate the p-value, *, $p < 0.05$; **, $p < 0.01$; ***, $p < 0.001$; ****, $p < 0.0001$.

Next, we determined if the altered Fzo1 levels are ubiquitinated as ubiquitin conjugation on Fzo1 can contribute to fusion or direct it towards proteolysis. The fusogenic ability depends on the ubiquitination status of the Fzo1 protein, dictated by the activity of Ubp2 [32]. To further investigate the ubiquitination status of the strains, we tagged Fzo1 C-terminally using hemagglutinin (HA) tag. The strains were then subjected to immunoprecipitation and pulled down using HA-specific antibodies. The pull-down fraction was subjected to immunoblotting and probed with an anti-ubiquitin antibody. We observed that the absence of Ubp2 leads to increased ubiquitination of Fzo1 as compared to WT (Fig 4E). However, Δ31Δ34 exhibits a marked decrease in the ubiquitination status of Fzo1, corroborating with its increased stability in the absence of Hsp31 paralogs. Interestingly, there was a significant restoration in the levels of ubiquitinated Fzo1 in Δ31Δ34Δubp2 as it is comparable to WT (Fig 4E and 4F). These findings highlight that the ubiquitination status of Fzo1 dictates its stability and turnover kinetics across the strains.

## 2.4 Ubp2 deletion restores cell cycle progression in cells lacking Hsp31 paralogs

Mitochondrial structure transition is integral for the cells to advance into subsequent cell cycle phases [43–45]. Previous studies indicate that cells lacking Hsp31 paralogs switch to hyperfused mitochondrial structures as an adaptive strategy to ameliorate the increased ROS levels. At the same time, this adaptive strategy is detrimental to cell cycle progression, as observed in the Δ31Δ34 strain [15]. Therefore, we hypothesized that remission of hyperfused mitochondrial structures in Δ31Δ34Δubp2 would restore the cell cycle progression.

To test this hypothesis, the strains were arrested at the G1 phase using alpha factor and synchronously released in dextrose media. Subsequently, the cells were fixed and subjected to flow cytometric analysis. Upon deletion of Ubp2, the cells exhibited no delay in cell cycle progression and resembled WT (Fig 5). The loss of Ubp2 does not cause perturbation in a cell cycle, as the mitochondrial distribution remains unaffected. However, the Δ31Δ34 strain exhibited slower progression as more than 50% of the cells continued to be at the G2/M phase (from t = 80 min to t = 100 min), per the presence of hyperfused mitochondrial structures [15] (Figs 5A and S14). Interestingly, restoration of mitochondrial integrity in Δ31Δ34Δubp2 led to the usual progression in cell cycle and is comparable to the WT. Taken together, the results indicate that remission of mitochondrial structures upon deletion of Ubp2 in the absence of Hsp31 paralogs enables the cells to undergo unimpeded progression through the cell cycle.

The progression of the normal cell cycle is well correlated with cell size and growth. Hence, we determined the change in cell size. Microscopic analysis revealed that Δ31Δ34 has cells with an average cell size of ~9 µm exhibiting increased cell size than WT (average cell size of 4–6 µm) and, in the stationary phase, exhibit pseudohyphal characteristics (Fig 5B and 5C) [15]. On the other hand, Δubp2 cells exhibit normal cell size compared to WT. Interestingly, Δ31Δ34Δubp2 has reversed the cell population with a size similar to the WT (Fig 5B and 5C). Thus, balanced mitochondrial dynamics are integral in maintaining cell cycle and cell size.

## 2.5 Ubp2 regulates basal ROS levels in the absence of Hsp31 paralogs

Active mitochondrial respiration contributes significantly to the pool of cellular basal ROS levels. Hence, we probed for the basal levels of ROS across the strains. The yeast cells grown in dextrose-containing media were treated with ROS sensing dye, $H_2$DCFDA (2',7',-dichlorodihydrofluorescein diacetate), and subjected to flow cytometric and microscopic analysis to measure the total cellular ROS levels. The flow cytometry data reveals that Δubp2 has basal ROS levels comparable to WT. A significant increase in the basal ROS levels was observed in Δ31Δ34, consistent with the previous findings (Fig 6A) [15]. Interestingly, the basal ROS levels were significantly restored in Δ31Δ34Δubp2 and the levels comparable to WT (Fig 6A). These findings were further confirmed by microscopic analysis with $H_2$DCFDA dye staining. Like flow cytometric data, a significant increment in green fluorescence intensity was observed for Δ31Δ34 strain. On the other hand, the green fluorescence was significantly restored in Δ31Δ34Δubp2 strains, similar to WT and Δubp2 strains (Fig 6B). These results were further validated by overexpressing Ubp2 in the background of Δ31Δ34Δubp2 and consistent with the findings, we found a significant increment

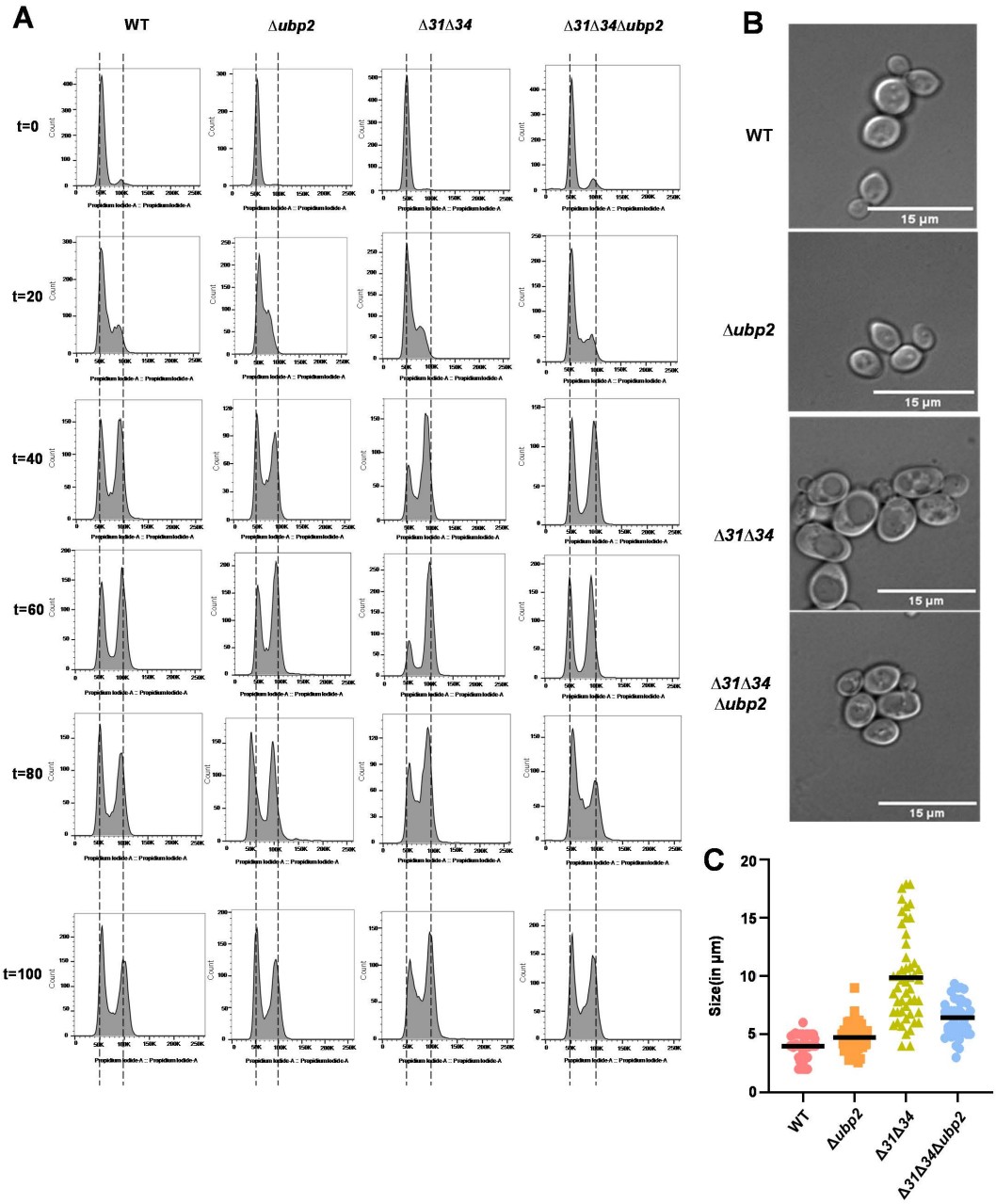

**Fig 5. Cell cycle is restored upon deletion of Ubp2 cells lacking Hsp31 paralogs. (A)** Cell cycle analysis**.** The restoration of cell cycle was analysed using PI staining after synchronizing the strains using alpha factor. Release after G1-arrest was compared at the indicated time points. **(B,C)** Assessment of cell morphology and size. The cell morphology and size in the indicated strains were analysed by microscopy **(B)**. Scale bar (15 μm). The relative differences in cell size in the yeast strains (WT, Δ31Δ34, and Δ31Δ34Δubp2 cells) were quantified using ImageJ **(C)**. The values were plotted using GraphPad Prism 5.0. One-way ANOVA with Tukey's multiple comparison test was used for significance analysis. Error bars represent the standard deviation in median values from 3 biological replicates. Asterisks indicate the p-value, *, $p < 0.05$; **, $p < 0.01$; ***, $p < 0.001$; ****, $p < 0.0001$.

in the basal cellular ROS levels indicated by increased staining, as compared to WT (S7B and S7C Fig). Interestingly, overexpression of the catalytically inactive mutant of Ubp2 did not increase the basal ROS levels, which was equivalent to WT and Δ31Δ34Δubp2 (S9 Fig);suggesting the role of cysteine dependent function of Ubp2 in regulating the redox status.

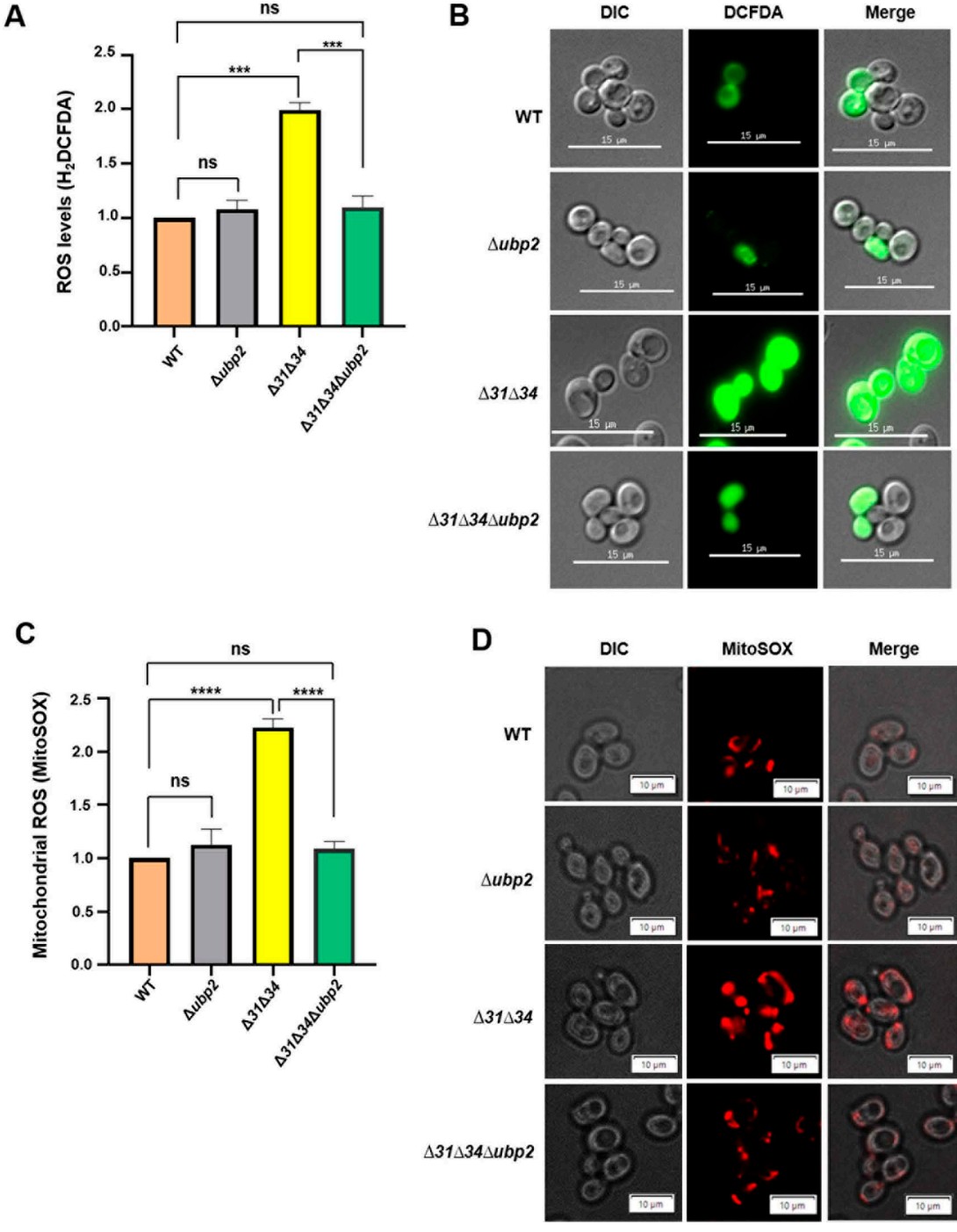

**Fig 6. Basal ROS levels are restored upon restoration of mitochondrial dynamics. (A,B)** Evaluation of basal ROS levels by $H_2$DCFDA. Total ROS levels were measured in indicated strains by staining the cells with $H_2$DCFDA and subjected to flow cytometric analysis **(A)**. Similarly, the cells from indicated strains were stained with $H_2$DCFDA and subjected to microscopic analysis **(B).** Scale bar (15 μm). **(C,D)** Evaluation of basal mitochondrial ROS. The mitochondrial ROS lelevls in indicated strains were determined by MitoSOX dye using flow cytometric analysis **(C).** Similarly, the mitochondrial ROS lelevls were probed by MitoSOX dye using microscopic analysis **(D)**. Scale bar (10 μm). One-way ANOVA with Tukey's multiple comparison test was used for significance analysis. Error bars represent the standard deviation in median values from 3 biological replicates. Asterisks indicate the p-value, *, $p < 0.05$; **, $p < 0.01$; ***, $p < 0.001$; ****, $p < 0.0001$.

The basal mitochondrial specific ROS was probed using mitochondrial specific dye, MitoSOX. The cells grown in dextrose-containing media were subjected to flow cytometry and microscopic analysis. The flow cytometric data revealed a non-significant difference in basal mitochondrial ROS in Δ*ubp2* strain, which is comparable to WT. However, a two-fold increment in mitochondrial ROS was observed in Δ*31*Δ*34* (Fig 6C). Interestingly, the mitochondrial ROS levels of Δ*31*Δ*34*Δ*ubp2* were restored to WT (Fig 6C). To substantiate the flow cytometry results, the microscopic analysis was performed using MitoSOX dye. The total mitochondrial ROS levels in WT and Δ*ubp2* were comparable (Fig 6D, *compare panels 1 and 2*). A significant increment in basal mitochondrial ROS in Δ*31*Δ*34* was evident upon the microscopic analysis (Fig 6D, *compare panels 1 and 3*). Consistent with the above findings, Δ*31*Δ*34*Δ*ubp2* exhibits a significant restoration of mitochondrial ROS distribution compared to WT (Fig 6D, *compare panels 1 and 4*), which correlates well with the reversal of mitochondrial integrity and functions. Overexpression of Ubp2 in the background of Δ*31*Δ*34*Δ*ubp2* leads to increment of the basal mitochondrial ROS levels as indicated by MitoSOX staining (S8 Fig). However, the cysteine mutant of Ubp2 was ineffective in increasing mitochondrial ROS upon its overexpression in Δ*31*Δ*34*Δ*ubp2* (S8 Fig).

## 2.6 Ubp2 modulates oxidative stress sensitivity in GSH dependent manner

Our experiments provide compelling evidence highlighting the significance of balanced mitochondrial dynamics in maintaining normal cellular growth. The findings also suggest that the mitochondrial integrity is restored in Δ*31*Δ*34*Δ*ubp2* strain due to the suppression of basal ROS levels. As mitochondrial health is directly correlated to restoration in the basal and mitochondrial ROS, we further probed for the response of the strains in the presence of extraneous oxidative stress. To test, the yeast cells from the mid-log phase were subjected to 1 mM $H_2O_2$ treatment, followed by growth analysis by spot test. The results reveal that Δ*ubp2* is not sensitive to extraneous oxidative stress as the growth is comparable to WT. On the other hand, Δ*31*Δ*34* showed maximum sensitivity to oxidative stress, consistent with previous findings [15]. Strikingly, the Δ*31*Δ*34*Δ*ubp2* strain showed partial restoration of sensitivity to extraneous stress (Fig 7A). In addition, Ubp2 was overexpressed in Δ*31*Δ*34*Δ*ubp2*, and the strains were subjected to phenotypic analysis upon $H_2O_2$ treatment. The growth assay suggests that upon complementation, the phenotype is reversed, whereby Δ*31*Δ*34*Δ*ubp2*/Ubp2(↑) is more sensitive to stress (S7A Fig). Interestingly, overexpression of the cysteine mutant of Ubp2 does not sensitise Δ*31*Δ*34*Δ*ubp2* cells to $H_2O_2$ stress and phenocopies Δ*31*Δ*34*Δ*ubp2* (S10 Fig).

Our results indicate that cross-talk between Ubp2 and Hsp31 paralogs regulates mitochondrial dynamics via Fzo1 and contributes to oxidative stress response. The glutathione pool is a crucial indicator of oxidative stress, wherein the reduced form of glutathione (GSH) acts as a ROS scavenger and mitigates oxidative stress. GSSG is the oxidized form of glutathione, which increases significantly in conditions of oxidative stress to cells. Interestingly, the GSH and GSSG ratio is also a crucial indicator of mitochondrial fusion, whereby GSSG mediates the stability of mitofusins and promotes mitochondrial fusion in conditions of oxidative stress [46]. As our results indicate that mitochondrial remodeling via Fzo1 is a consequence of oxidative stress response, we further probed for the GSH levels using monochlorobimaine (MCB). Monochlorobimine is a cell-permeable nonfluorescent probe that shows fluorescence only when it reacts with the reduced glutathione pool in cells. Cells in the mid-log phase were treated with MCB and subjected to microscopy and flow cytometric analysis. It is evident from microscopic analysis that Δ*ubp2* does not show any difference in MCB staining as the fluorescence intensity is comparable to WT (Fig 7B). This corroborates the fact that Fzo1 stability is altered without affecting the redox parameters in the absence of Ubp2. Interestingly, Δ*31*Δ*34* exhibits a significant decrease in GSH levels, as indicated by the minimal MCB fluorescence in cells lacking Hsp31 paralogs (Fig 7B). Most interestingly, Δ*31*Δ*34*Δ*ubp2* exhibit restored levels of GSH, thereby restoring redox parameters (Fig 7B).

Furthermore, the flow cytometric analysis indicated a similar pattern of MCB uptake across the indicated strains. GSH levels remain unaltered in Δ*ubp2* and are comparable to WT (Fig 7C). In line with the microscopic results, flow cytometric analysis showed a significant decrement in GSH levels in Δ*31*Δ*34* strain. At the same time, Δ*31*Δ*34*Δ*ubp2* exhibits restored GSH levels comparable to WT (Fig 7B).The complementation study further validated these results, wherein

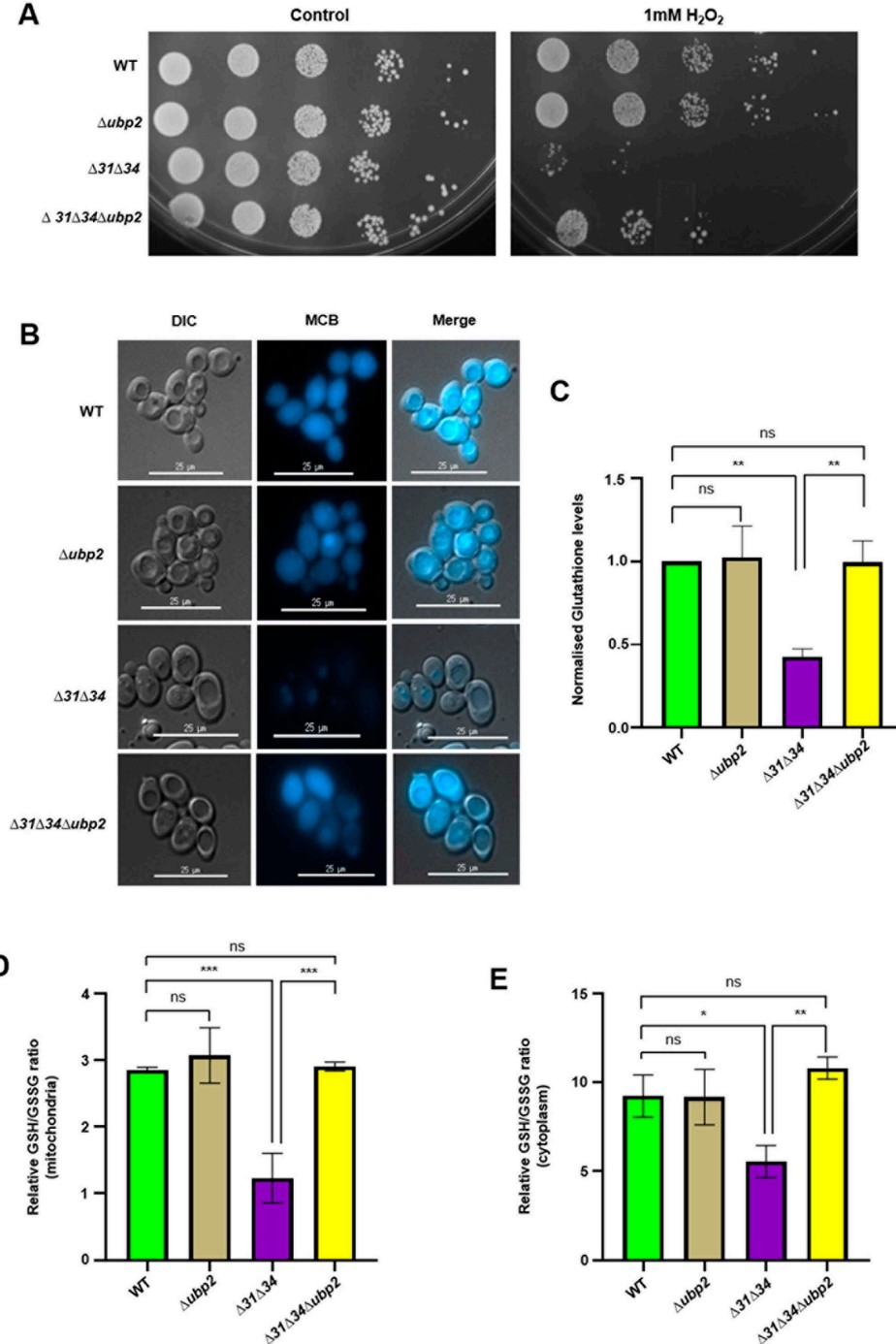

**Fig 7. Loss of Ubp2 provides partial oxidative stress resistance in the absence of Hsp31 paralogs. (A)** Growth phenotype assessment. Yeast strains were grown till mid-log phase and were subjected to $H_2O_2$ stress. **(B,C)** Evaluation of Glutathione levels using flow by microscopy and cytometry. The GSH levels were determined by monochlorobiamine staining and subjected to microscopy **(B)**, Scale bar (25 µm). The normalised GSH levels were probed using flow cytometry **(C)** by monochlorobiamine staining. **(D, E)** Assessment of relative GSH/GSSG ratio in the mitochondrial fraction **(D)** and cytoplasmic fraction **(E)** of the indicated strains, using Glutathione assay kit. One-way ANOVA with Tukey's multiple comparison test was used for significance analysis. Error bars represent the standard deviation in median values from 3 biological replicates. Asterisks indicate the p-value, *, $p < 0.05$; **, $p < 0.01$; ***, $p < 0.001$; ****, $p < 0.0001$.

*Δ31Δ34Δubp2/Ubp2↑* exhibits lesser GSH level and is comparable to *Δ31Δ34* (S11 Fig). Additionally, the GSH and GSSG levels were measured in the mitochondrial and cytoplasmic compartments. The GSH/GSSG ratio decreased significantly in *Δ31Δ34* in the mitochondria (Fig 7D) and cytoplasm (Fig 7E), indicating an increment in the oxidized glutathione levels. Opposingly, *Δ31Δ34Δubp2* has a GSH/GSSG ratio comparable to WT, indicating restored redox status (Fig 7D and 7E). As the GSH and GSSG levels were perturbed in *Δ31Δ34*, we reinstated the redox status by treating the *Δ31Δ34* cells with N-acetyl cysteine (NAC). We found that the mitochondrial morphology and total mass were reinstated and resembled WT (S17A and S17C Fig). Moreover, the reversal of mitochondrial phenotype in NAC-treated *Δ31Δ34* cells is due to a decrease in Fzo1 steady state levels (S17B Fig), thereby highlighting the role of Hsp31 paralogs in regulating mitochondrial dynamics in a redox-dependent manner.

In summary, our observations firmly establish a cross-talk between Ubp2 and Hsp31 paralogs in maintaining mitochondrial health in normal physiological conditions and under oxidative stress.

## 3 Discussion

Balanced mitochondrial dynamics is one of the critical aspects that assist cells in maintaining a healthy mitochondrial pool. An imbalance in mitochondrial dynamics is one of the common denominators in the progression of several pathological disorders, such as cancers and neurodegenerative disorders [47,48]. One of the well-studied neurodegenerative diseases, such as Parkinson's disorder (PD), is caused by genetic mutations in *the PARK7* gene, which encodes for DJ-1 [17,20,49]. Early evidence suggests that the DJ-1 maintains mitochondrial homeostasis by modulating the conserved mitochondrial dynamic proteins [15,26,50–52].As the members of the DJ-1 family are ubiquitous, they perform many similar functions throughout the phylogeny. *S. cerevisiae* consists of four paralogs belonging to the DJ-1 family, which modulate mitochondrial integrity and oxidative stress response [14,15]. Although, the paralogs (Hsp31, Hsp32, Hsp33 and Hsp34) are involved cytoprotection, only Hsp31 and Hsp34 paralogs work in conjunction to maintain mitochondrial dynamics, as their combinatorial deletion perturbs mitochondrial morphology [15,16]. Besides, the Hsp34 is much more divergent in sequence than the Hsp32 and Hsp33 paralogs. Therefore, Hsp34 has better evolved vital functions in conjunction with Hsp31 in maintaining overall cellular and organellar health. The perturbed mitochondrial dynamics in the absence of Hsp31 paralogs are due to altered expression of mitochondrial fusion protein, particularly Fzo1 (Fig 8). However, the molecular mechanism of how DJ-1 paralogs modulate mitochondrial dynamics through fusion protein is still elusive across phylogeny.

Yeast Fzo1 belongs to a conserved class of mitofusins essential for the fusion of the outer mitochondrial membrane. Several studies indicate that the ubiquitination of mitofusin is altered in response to stress and is critical for regulating mitophagy, apoptosis, and maintenance of ER-mitochondrial contacts [53–55]. Taken together, the ubiquitination of mitofusins is pivotal in regulating mitochondrial dynamics and cellular adaptability to metabolic cues [38,56]. In *S. cerevisiae*, the fusogenic activity of Fzo1 is finely regulated by conserved Ubp2, which stabilizes the 'pro-fusion' ubiquitin chain on Fzo1 and prevents its degradation (Fig 8) [32,39]. Therefore, yeast cells lacking Ubp2 exhibit growth sensitivity on a non-fermentable carbon source at 37°C due to the accumulation of fragmented, non-functional mitochondria, thus highlighting its role in regulating mitochondrial health in response to heat stress.

In line with this evidence, the present study establishes first-time a genetic cross-talk between Ubp2, a key mediator in modulating DJ-1-dependent mitochondrial dynamics in eukaryotes. The cells lacking Ubp2 in the absence of both Hsp31 & 34 paralogs fully restore the healthy pool of mitochondria, further substantiating a functional dependency between DJ-paralogs and Ubp2 in maintaining mitochondrial health. The expression of Ubp2 is variable in different phases of cellular growth. On the contrary, Ubp2 levels were elevated but remained unaltered in cells lacking Hsp31 paralogs in various growth phases, suggesting that DJ-1 paralogs regulate the growth phase-dependent expression of Ubp2. Moreover, the increased expression of Ubp2 contributes to the increased stability of Fzo1 in *Δ31Δ34 cells*, thus mediating elevated events of mitochondrial fusion (Fig 8). Differential expression of Ubp2 in the absence of Hsp31 paralogs further substantiates their role in reprogramming of the cellular proteome [56 ].

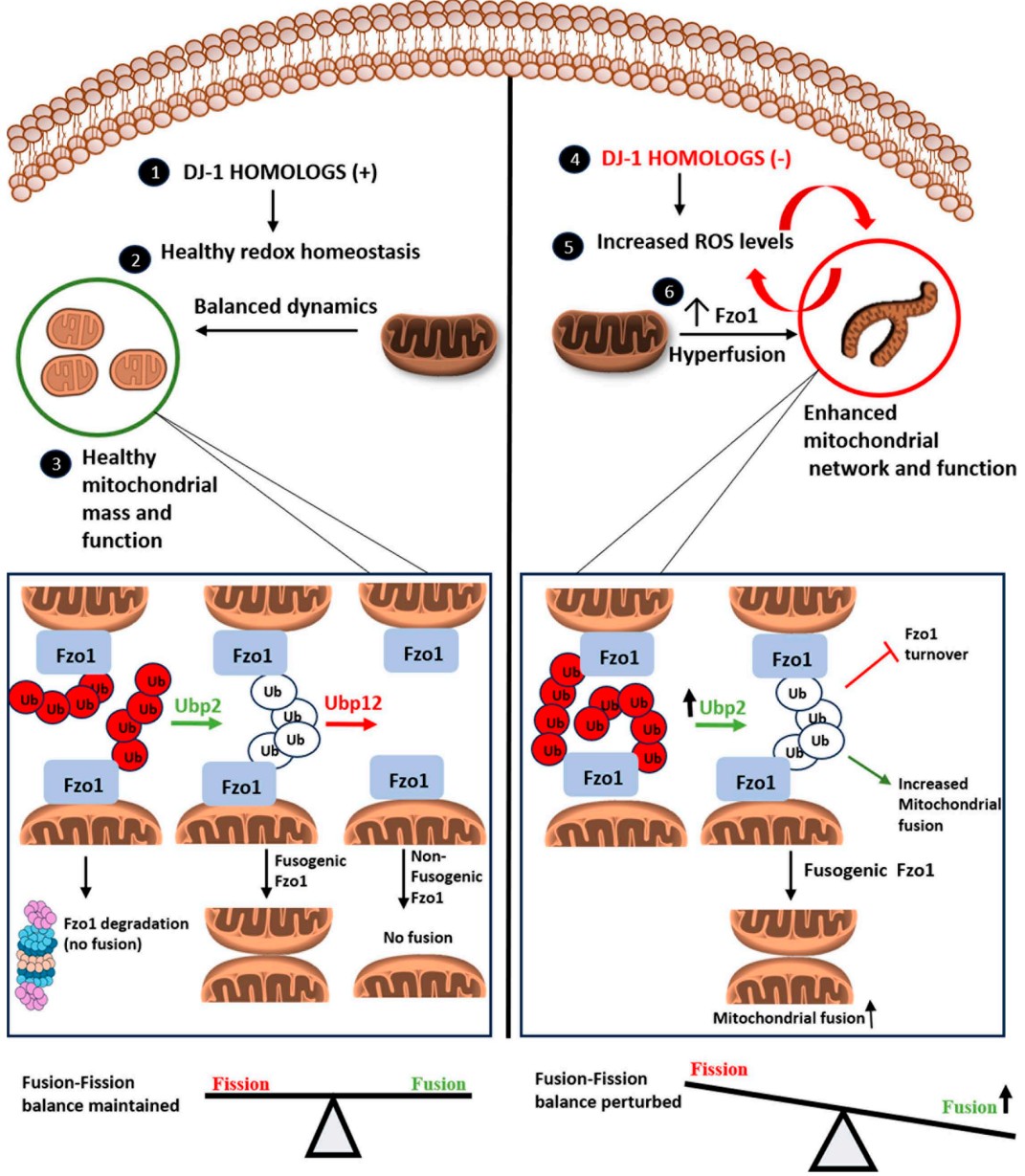

**Fig 8. Model depicting the role of Ubp2 in regulating mitochondrial dynamics in cells lacking Hsp31 paralogs.** (1) In healthy cells (left panel), DJ-1 homologs maintain optimum redox homeostasis by regulating balanced fusion and fission events. (2) The fusion-fission balance is crucial for maintaining healthy mitochondrial mass and function. (3) The ubiquitination status of Fzo1 dictates either fusion or degradation. Ubp2 mediates fusion by editing the ubiquitin chain on Fzo1, promoting mitochondrial fusion. Ubp12 opposes the function of Ubp2 by antagonizing fusion. (4) In cells lacking DJ-1 paralogs (right panel), the redox status is perturbed due to increased basal ROS levels. (5) The elevated ROS is due to an increase in the functional hyperfused mitochondrial structures, which in turn is used by the cells as an adaptive strategy. (6) The imbalance in fission-fusion dynamics in cells lacking DJ-1 paralogs is attributed to increased Fzo1 levels. The ubiquitination status of Fzo1 is altered due to increased expression of Ubp2, leading to enhanced fusion events and hence perturbation in the mitochondrial dynamics. The model was made using modified mitochondria icons from https://bioicons.com/ (simple_mitochondria_network icon by Marnie-Maddock https://github.com/MarnieMaddock is licensed under CC-BY 4.0 Unported https://creativecommons.org/licenses/by/4.0/);(mitochondria-grey icon by DBCLS https://togotv.dbcls.jp/en/pics.html is licensed under CC-BY 4.0 Unported https://creativecommons.org/licenses/by/4.0/), (mitochondrium-yellow icon by Servier https://smart.servier.com/ is licensed under CC-BY 3.0 Unported https://creativecommons.org/licenses/by/3.0/) and the rest of the figure was drawn.

Recent studies elicit the role of Ubp2 in protecting Fzo1 from proteasomal degradation by cleaving the ubiquitylated forms of Fzo1, which repress mitochondrial fusion (Fig 8) [32,38]. Nevertheless, the increased stability of Fzo1 in the absence of Hsp31 paralogs is due to reduced ubiquitination of Fzo1. On the contrary, cells lacking Ubp2 exhibit reduced stability of Fzo1 due to the elevated levels of ubiquitinated moieties (Fig 8). At the same time, the stability and ubiquitination status of Fzo1 in Δ31Δ34Δubp2 cells is reinstated to the WT level, thus validating that Ubp2 functions in conjunction with Hsp31 paralogs and affects the mitochondrial dynamics exclusively via Fzo1 (Fig 8).

Mitochondrial structures, plasticity, and transport are crucial determinants of cell cycle progression [43–45]. The cells lacking both Hsp31 & 34, which exhibit hyperfused mitochondrial structures, cause a delay in the G2/M phase of the cell cycle. However, restoring mitochondrial dynamicity in the absence of Ubp2 in cells lacking DJ-1 paralogs reverses the delay in G2/M progression and resumes normal cell cycle progression. This further underscores the importance of functional interaction between Hsp31 paralogs and Ubp2 in maintaining the cell cycle by regulating mitochondrial integrity.

Mitochondrial structure and function are closely intertwined and critical for regulating redox homeostasis. Hsp31 paralogs are known to protect cells from carbonyl stress, thereby alleviating the concomitant formation of ROS species. Hence, cells lacking Hsp31 paralogs exhibit elevated levels of basal ROS [16]. This is mainly due to cells lacking Hsp31 paralogs exhibiting enhanced functional networked mitochondrial structures as an adaptive strategy, thereby increasing basal ROS levels [15]. However, this adaptive strategy is detrimental to the cells due to their susceptibility to oxidative stress. Upon deletion of Ubp2 together with the Hsp31 paralogs, mitochondrial hyperfusion is suppressed and the GSH/GSSG ratio is restored, thereby reinstating the basal redox homeostasis (Fig 8). Hence, the cells lacking Ubp2 and Hsp31 paralogs are more resistant to extraneous oxidative stress. Further, Ubp2 is involved in oxidative stress response, whereby Ubp2 is inactivated reversibly under $H_2O_2$ stress [33]. This further underlines the crucial role of Ubp2 in regulating stress response and bolsters the link between Ubp2 and redox-sensitive Hsp31 paralogs.

Our study provides crucial findings that elucidate the cross-talk between Ubp2 and DJ-1 paralogs for the first time in regulating mitochondrial health. In humans, DJ-1 is a neuroprotective protein orthologous to yeast Hsp31 paralogs and is crucial for maintaining a healthy cellular environment. The familial mutations in the human DJ-1 (PARK7) protein cause severe pathophysiological manifestations leading to neurodegeneration in PD. Notably, the mutations disrupt mitochondrial structure and function, eventually affecting the oxidative stress sensitivity of the cells. Owing to the significance of DJ-1 in maintaining mitochondrial health, our study unravels the mechanism by which DJ-1 paralogs regulate mitochondrial dynamics in *Saccharomyces cerevisiae*. The knowledge of molecular intricacies that integrate functions of deubiquitinase and DJ-1 via mitochondrial regulation can provide better insights into mechanistic understanding of Parkinson's disease progression.

## 4 Materials and methods

### 4.1 Generation of yeast strains and plasmid construction

The yeast strains utilized in this study are mentioned in S1 Table. Deletions were done using homologous recombination in the background of BY4741 (MATα his3Δ1 leu2Δ0 met15Δ0 ura3Δ0). Strains with deletions of Hsp31 paralogs (Δ31, Δ34, and Δ31Δ34) were generated previously in the lab [14]. Ubp2 was knocked out by generating an appropriate DNA cassette comprising regions homologous to 5' and 3' UTR of the gene *UBP2* using primer sets P5/P6, with *nat1* as the selection marker. The cassette was transformed in WT, Δ31, Δ34, and Δ31Δ34 and consequently, Δubp2, Δ31Δubp2, Δ34Δubp2, Δ31Δ34Δubp2 strains were generated. Subsequently, the deletion strains were confirmed using primers P7 and P8.

For cell cycle analysis, Δbar1 and Δ31Δ34Δbar1 were generated previously in the lab [15], and Δ31Δ34Δubp2Δbar1 was generated using primers P5 and P6. All the deletions were confirmed using the primers P7/P8. Ubp2, Fzo1, and Dnm1 were HA-tagged genomically at the C-terminus by transforming the indicated strains with DNA cassettes encoding for *hphNT1*resistance gene with P9/P10, P1/P2, and P12/P13, respectively.

*UBP2* with HA-tag at C-terminus was cloned into pRS415 plasmid with TEF promoter for complementation studies. The constructed plasmid was transformed into Δ*ubp2* and Δ*31Δ34Δubp2*. For mitochondria visualization with a microscope, the strains were transformed with a previously constructed plasmid expressing MTS-mCherry under the influence of TEF promoter [14].

## 4.2 Western blot analysis

The indicated strains were subjected to growth until the log phase ($A_{600}$=0.6-0.8 OD) or until the stationary phase ($A_{600}$=1–2 O.D.) in necessary experiments. An equal O.D. ($A_{600}$=2.5 OD) was pelleted for the indicated strains and subjected to incubation with 10% TCA (trichloroacetic acid) at 4°C. It was followed by acetone washes and subjected to lysis by bead beating method. Acid-washed glass beads were added to the dried pellets along with 1X SDS dye (50 mM Tris-HCl, pH 6.8, 2% SDS, 0.1% bromophenol blue, 10% glycerol, and 100 mM β-mercaptoethanol). Consequently, the samples were heated at 92°C for 15 min, followed by a spin at 14,000 rpm at room temperature (23–25°C). The supernatant was loaded onto different percentages of SDS-polyacrylamide gels depending on the molecular weights of the protein to be separated and probed. The total protein content in each strain was quantified by staining the polyacrylamide gel and normalized for equal loading. The samples were transferred to an activated PVDF membrane for immunoblotting analysis. The blots were first blocked in 0.1% TBST skim milk, followed by multiple washes with 0.1% TBST. The blots were incubated with the respective primary and secondary antibodies (mentioned individually and in the S2 Table), followed by probing with luminol solution (BioRad).

## 4.3 Microscopic analysis

The mitochondrial morphology was analyzed for indicated strains by transforming the mTS-mCherry construct. Due to the mitochondrial targeting signal sequence, the mitochondria were decorated explicitly with mCherry and visualized using a Delta Vision Elite microscope at 100X magnification. The transformed cells were collected from the mid-log phase and spread on agar padding. These non-fixed cells were subjected to microscopic analysis.

To measure the ROS with specific-dyes ($H_2$DCFDA and MitoSox), cells from the mid-log phase were subjected to staining with the respective dyes in the dark. After incubation of the cells with dye for 30 min, the cells were washed with sterile PBS. The dye-treated cells were spread on agarose padding and subjected to microscopy analysis at 100x magnification.

## 4.5 Quantification of mitochondrial network morphology

ImageJ was used to measure different mitochondrial morphologies across the strains. Maximum intensity projections of all the fields were created, following which 'thresholding' was done to remove the background. Particles were measured by setting a scale based on the pixels by using the 'Measure particle' function. The particles were then analyzed based on a range using the 'analyze particle' function. Cells were counted per field, and the mitochondrial morphologies were scored based on the dimensions measured.

## 4.6 Measurement of total and functional mitochondrial mass

Total mitochondrial mass was measured by staining the cells with cardiolipin-specific dye called nonyl acridine orange (NAO). Cells from the mid-log phase ($A_{600}$=0.6 OD) were treated with NAO for 30 min (in the dark) at 30°C at 900 rpm. To understand the role of redox status in regulating mitochondrial mass, cells were grown overnight with 5 mM NAC and subjected to flow cytometry after staining with NAO. Similarly, cells from the mid-log phase were treated with a potentiometric sensitive dye called JC-1 to measure the functional mitochondrial mass. Subsequently, the cells were washed in sterile PBS and subjected to flow cytometric analysis (BD FACS Verse).

## 4.7 Pull down analysis

The ubiquitination status of Fzo1 was determined by the pull-down assay performed according to a previously established protocol [57]. Briefly, 200 mL of culture from the mid-log phase ($A_{600}$=0.6 OD) was pelleted down and suspended

in ice-cold immunoprecipitation (IP) buffer (1 M Tris pH = 7.5, 5 M NaCl, 0.5 M EDTA). The slurry was subjected to bead beating (1 min, four cycles) with intermittent incubation on ice. The lysed cells were centrifuged at 16000 rpm for 5 min at 4°C. The supernatant (containing the cytosolic fraction) was discarded, and the pellet (containing all the membrane fractions) was subjected to solubilization using IP buffer containing 0.5% Nonidet P-40 (NP-40). The membrane fraction was solubilized using a nutator at 4°C for 2h. Meanwhile, Protein G beads (Protein G Sepharose 4 Fast Flow beads, Catalogue no: 17061801, Cytiva) were equilibrated with IP buffer and were incubated with anti-HA antibody (Monoclonal anti-HA tag antibody, Catalogue no: SAB2702217, Sigma Aldrich) at 10 rpm, 4°C for 2h. Consequently, the antibody equilibrated agarose beads were incubated with the solubilized membrane fraction for binding (10 rpm, 4°C, 12h). The protein bound beads was eluted with loading dye and separated on 8% SDS-PAGE gel and subjected to blotting on PVDF-memebrane (37.5:1 ratio of acrylamide: polyacrylamide) [57]. The blots were incubated with either with anti-HA (Monoclonal anti-HA tag antibody, Catalogue no: SAB2702217, Sigma Aldrich) and anti-ubiquitin antibody (Ubiquitin antibody, Catalogue no: NB300–130, Novus Biologicals). Anti-mouse secondary antibody (Amersham ECL Mouse IgG, Catalogue no: NA931V, Cytiva) was used for probing blots with anti-HA antibody and anti-ubiquitin antibody. The blots were consequently probed using luminol solution (BioRad).

## 4.8 Estimation of ROS levels

To probe for the basal cellular and mitochondria specific ROS, cells from mid-log phase were treated with $H_2DCFDA$ and MitoSox respectively. The dye treated cells were subjected to flow cytometry and microscopic analysis. To measure the ROS levels after extraneous stress, cells from the mid-log phase were treated with 1 mM $H_2O_2$. The treated cells were washed with 1X PBS and stained with the respective dyes.

## 4.9 Mitophagy induction

All the indicated strains were tagged with GFP at the C-terminus of mitochondrial OM45. The strains were then subjected to growth in YP-dextrose media till the mid-log phase. These cells were shifted to grow in YP-glycerol media for the indicated time points at the permissive temperature of 30°C, thereby inducing mitophagy. Lysates were collected and subjected to Western blotting. The blots were probed using anti-GFP antibody (Anti-Green Fluorescent Protein (GFP) antibody, Catalogue no: G1546, Sigma-Aldrich), followed by probing with anti-mouse secondary antibody(Amersham ECL Mouse IgG, Catalogue no: NA931V, Cytiva). The densitometric intensities of processed GFP and OM45-GFP were quantified and normalized to the corresponding loading control. Consequently, the ratio of processed GFP to total GFP (processed GFP + OM45-GFP) was calculated and plotted. Furthermore, cells undergoing mitophagic induction were subjected to microscopic analysis, and the vacuolar localization of processed OM45-GFP was detected.

## 5 Visualization of vacuole for mitophagy

Following 54h of growth in YP-glycerol, all the indicated OM45 GFP-tagged strains were subjected to staining with FM4–64 dye (0.5 μg/ml) in PBS. After dye treatment for 2h at 30°C, fresh YP-glycerol was added and incubated for 30 min at 30°C, 900 rpm. Consequently, the media was removed, cells were suspended in PBS, and subjected to microscopy as described previously.

## 5.1 Estimation of glutathione

Mitochondrial and cytoplasmic fractions were isolated and seperated from the indicated strains,as per the protocol previously published [58]. The purity of the mitochondrial fractions was checked by probing the mitochondrial extracts with anti-Tim44(expressed in mitochondrial inner membrane) antibody (Gifted, Elizabeth A Craig lab,University of Wisconsin-Madison)along with anti-Pgk1 antibody (Gifted, Pundi N Rangarajan lab, Indian Institute of Science) [59,60]. The total glutathione and GSSG levels were measured using the glutathione GSH/GSSG assay kit (Sigma, Catalogue no.

MAK440) as per the indicated protocols. Mitochondrial isolates (100 μg protein) and cytoplasm (100 μg protein) were incubated with the reagents, and absorbance was recorded at the wavelength of 460nm. All the values were then calculated based on the standard curve, and the ratio was plotted.

**5.2 Statistical analysis**

All statistical analysis was performed using GraphPad Prism 6.0 software. Error bars represent Standard deviation (S.D) derived from at least three biological replicates.The respective statistical test performed has been performed for the experiments represented Asterisks used in the figures represent the following significance values: *, $p \le 0.05$; **, $p \le 0.01$; ***, $p \le 0.001$; ****, $p \le 0.0001$; ns, not significant.

## Supporting information

**S1 Fig. Growth kinetics. (A)** WT and *Δ31Δ34 strains* containing HA-tagged Ubp2 at the C-terminus were grown in dextrose-containing media for indicated time intervals. The early-log phase (6h) and late-log phase (12h) are highlighted. **(B)** The cell lysates prepared from strains at indicated time intervals were subjected to Ubp2 expression analysis by Western blotting. **(C)** The blots were quantified by densitometry, and the change in the expression levels of Ubp2-HA was represented. One-way ANOVA with Tukey's multiple comparison test was used for significance analysis. Error bars represent the standard deviation in median values from 3 biological replicates. Asterisks indicate the p-value, *, $p < 0.05$; **, $p < 0.01$; ***, $p < 0.001$; ****, $p < 0.0001$.
(TIFF)

**S2 Fig. Expression levels of Ubp2-HA upon complementation.** The equivalent amount of cells from indicated strains used in the complementation spot assay were lysed and subjected to immunoblotting to check for equal expression.
(TIFF)

**S3 Fig. Ubp2 deletion in *Δ31* and *Δ34* leads to the accumulation of fragmented mitochondria.** Mitochondrial morphology in the indicated strains transformed with mTS-mCherry was analyzed by microscopy. Scale bar (10 μm).
(TIFF)

**S4 Fig. Assessment of mitophagy by OM45-GFP processing. (A, B)** The indicated strains were subjected to mitophagy induction, and lysates were collected at different time points. Western blotting was performed to probe the processed GFP **(A)** and quantified from the triplicates by densitometry. Graph depicting the ratio of free GFP by the total GFP (processed and unprocessed) quantified **(B)**.
(TIFF)

**S5 Fig. Ubp2 overexpression in WT stabilizes Fzo1. (A)** Growth phenotype assessment. The indicated yeast strains were allowed to grow up to the mid-log phase in SC dextrose broth at 30°C. Ten-fold serially diluted cells were spotted on the indicated media and incubated at permissive temperature (30°C) and non-permissive temperature (37°C). Images were captured at 72h for glycerol. **(B)** Assessment of mitochondrial integrity. Yeast strains expressing MTS-mCherry grown in SC Leu- dextrose till the mid-log phase were subjected to microscopic analysis to visualize mitochondria. Scale bar (10 μm). **(C)** Lysates prepared from the cells were analyzed for the differences in Fzo1 expression by Western blotting.
(TIFF)

**S6 Fig. Ubp2 overexpression in *Δ31Δ34Δubp2* reinstates Fzo1 expression and stability. (A)** Fzo1 steady-state levels were probed in the indicated strains by Western blotting using anti-Fzo1 antibody. **(B)** Cycloheximide Chase assay. The cells from yeast strains were subjected to treatment with cycloheximide for the indicated time points, and the lysates were subjected to western blotting using anti-Fzo1 antibody.
(TIF)

**S7 Fig. Ubp2 overexpression in *Δ31Δ34Δubp2* causes reversal of redox sensitivity. (A)** Phenotypic analysis upon treatment with $H_2O_2$. Cells were treated with 1 mM $H_2O_2$ and spotted in SC-Leu-Dextrose plates. Images were captured after 36 h. **(B)** $H_2DCFDA$ treatment was performed to analyse ROS levels by microscopy. Scale bar (15 μm).
(TIFF)

**S8 Fig. Microscopic analysis of mitochondrial basal ROS using MitoSOXupon Ubp2 overexpression.** Cells were subjected to staining with MitoSOX to estimate mitochondrial basal ROS upon overexpression of Ubp2 and its cysteine mutant (Ubp2$_{C745S}$). Scale bar (10 μm).
(TIFF)

**S9 Fig. Microscopic analysis of cellular basal ROS using $H_2DCFDA$ upon Ubp2 overexpression.** Overexpression of Ubp2 increases basal ROS levels,while the Ubp2$_{C745S}$ mutant does not affect the redox status.The indicated strains were stained with $H_2DCFDA$ and visualised by microscopy. Scale bar (15 μm).
(TIFF)

**S10 Fig. Phenotypic assessment in response to $H_2O_2$ stress.** The indicated strains were grown till mid-log phase and treated with 1 mM $H_2O_2$ followed by spot assay at 30°C.The expression levels of Ubp2$_{C745S}$ is comparable to Ubp2.
(TIFF)

**S11 Fig. Microscopic assessment of glutathione levels by monochloramine (MCB) staining.** The indicated strains were stained with MCB and subjected to microscopy.Scale bar (10 μm).
(TIFF)

**S12 Fig. Microscopic analysis of mitochondrial morphology upon Ubp2 overexpression.** Ovexpresssion of Ubp2 in *Δ31Δ34Δubp2* causes hyperfusion of mitochondria. The indicated strains were transformed with MTS-mCherry construct and analysed by microscopy. Scale bar (10 μm).
(TIF)

**S13 Fig. Assessment of total and functional mitochondrial mass upon Ubp2 overexpression. (A)** Cells grown until the mid-log phase were subjected to NAO staining and quantified using BD FACS verse instrument. **(B)** Evaluation of the functional mitochondrial mass by flow cytometry. Cells grown until the mid-log phase were subjected to TMRE staining, a potentiometric dye, and acquired using BD FACS Verse.
(TIFF)

**S14 Fig. Quantification of percentage of cells in different phases of cell cycle: Cells were subjected to G1 phase arrest by treatment with alpha fator and subsequently released into fresh media.** A graph was plotted depicting the percentage of cells at each time interval.
(TIFF)

**S15 Fig. Dnm1 levels remain unaltered.** WT, *Δubp2*, *Δ31Δ34*, *Δ31Δ34Δubp2* were tagged with HA at the C-terminus of Dnm1 and lysates were subjected to Western blotting to check for difference in expression.
(TIFF)

**S16 Fig. Phenotypic assessment in respiratory media upon Ubp2 overexpression.** The indicated strains were serially diluted and spotted on the indicated media across different temperatures.
(TIFF)

**S17 Fig. Assessment of mitochondrial integrity upon reversal of basal ROS by NAC treatment. (A)**The mitochondrial morphology of the indicated strains was visualised by utilising mts-mCherry constructs. Scale bar(10μm). **(B)**Fzo1

levels were probed by Western blotting. **(C)**The total mitochondrial mass was measured by NAO staining,followed by flow cytometry.
(TIF)

**S18 Fig. Cells in post diauxic shift exhibit stress resistance.** Phenotype assessment of cells from mid-log phase and post diauxic shift phase. The cells were treated with $H_2O_2$ and spot assay was performed.
(TIFF)

**S19 Fig. Growth phenotype assessment in *Δ31Δ34Δubp2* upon Hap4 overexpression.** The indicated yeast strains were allowed to grow up to mid-log phase in SC dextrose broth at 30°C. Ten-fold serially diluted cells were spotted on the indicated media and incubated at permissive temperature(30°C).
(TIFF)

**S1 Table. List of primers and strains used in the study.**
(DOCX)

**S2 Table. List of antibodies used in the study.**
(DOCX)

**S3 Table. List of reagents used in the study.**
(DOCX)

**S1 File.** Quantification for densitometric difference in Ubp2-HA expression in WT and Δ31Δ34 (corresponding to Fig 1C).
(XLSX)

**S2 File.** Percentage of cells with different mitochondrial morphologies, as depicted graphically (corresponding to Fig 2B).
(XLSX)

**S3 File.** Assessment of total mitochondrial mass by NAO staining (corresponding to **Fig 2C**).
(XLSX)

**S4 File.** Quantitation of mitochondrial membrane potential after JC-1 staining across the indicated strains (corresponding to **Fig 2D**).
(XLSX)

**S5 File.** Assessment of ATP levels across the indicated strains (corresponding to **Fig 2E**).
(XLSX)

**S6 File.** Assessment of GFP processing by measuring the ratio of processed and total GFP (corresponds to **Fig 3B**).
(XLSX)

**S7 File.** Measurement of steady state levels of Fzo1 (corresponding to **Fig 4B**).
(XLSX)

**S8 File.** Assessment of Fzo1 turnover kinetics by cycloheximide chase assay and immunoblotting (corresponding to **Fig 4D**).
(XLSX)

**S9 File.** Determining the ubiquitination status of Fzo1 after pull down and probing with anti-ubiquitin antibody (corresponding to **Fig 4F**).
(XLSX)

**S10 File.** Quantification for cell size of the strains indicated in Fig 5B (corresponding to Fig 5C).
(XLSX)

**S11 File.** Measurement of total ROS levels by H2DCFDA staining (corresponding to Fig 6A).
(XLSX)

**S12 File.** Measurement of mitochondrial ROS by MitoSox (corresponding to Fig 6B).
(XLSX)

**S13 File.** Assessment of glutathione levels by monochlorobiamine staining (corresponding to Fig 7B).
(XLSX)

**S14 File.** Relative GSH/GSSG ratio as measured in mitochondria isolated from the indicated strains (corresponding to Fig 7D).
(XLSX)

**S15 File.** Relative GSH/GSSG ratio as measured in cytoplasmic fractions of the indicated strains (corresponds to Fig 7E).
(XLSX)

**S16 File.** Measurement of growth kinetics (corresponding to S1A Fig).
(XLSX)

**S17 File.** Quantification of fold change in Ubp2-HA expresssion levels in early log and late log phases (corresponding to S1C Fig).
(XLSX)

**S18 File.** Percentage of cells with different mitochondrial morphologies, as depicted graphically (corresponding to S5B Fig).
(XLSX)

**S19 File.** Measurement of steady state levels of Fzo1 (corresponding to S6A Fig).
(XLSX)

**S20 File.** Assessment of Fzo1 turnover kinetics by cycloheximide chase assay and immunoblotting (corresponding to S6B Fig).
(XLSX)

**S21 File.** Measurement of total ROS levels by H2DCFDA staining (corresponding to S7C Fig).
(XLSX)

**S22 File.** Percentage of cells with different mitochondrial morphologies, as depicted graphically (corresponding to S12 Fig).
(XLSX)

**S23 File.** Measurement of total mitochondrial mass after Ubp2 overexpession (corresponding to S13A Fig).
(XLSX)

**S24 File.** Measurement of functional mitochondrial mass after Ubp2 overexpession (corresponding to S13B Fig).
(XLSX)

**S25 File.** Quantification of percentage of cells in different phases of cell cycle (corresponding to S14 Fig).
(XLSX)

**S26 File.** Measurement of steady state levels of Dnm1 (corresponding to S15 Fig).
(XLSX)

**S27 File.** Quantification of total mitochondrial mass by NAO (corresponding to S17C Fig).
(XLSX)

## Acknowledgments

Fzo1 antibody is a kind gift from Prof. Mafalda Escobar-Henriques, Centre for Molecular Medicine, Cologne, Germany. Poly ubiquitin-specific antibody is a kind gift from Prof. Utpal Nath, Indian Institute of Science, Bengaluru. Pgk1 is a kind gift from Prof. Pundi N Rangarajan, Indian Institute of Science, Bengaluru and Tim44 is a kind gift from Prof. Elizabeth A Craig, Department of Biochemistry, University of Wisconsin-Madison. We thank the Flow Cytometry facility of the Indian Institute of Science, Bengaluru. We thank Dr. Saravanan Palani, Indian Institute of Science, Bengaluru for the microscopy facility. FM4–64 dye is a kind gift from Prof. Ravi Manjithaya, Jawaharlal Nehru Centre for Advanced Scientific Research, Bengaluru. We are grateful to Dr. Rachayeeta Deb (DBT-RA, Indian Institute of Science, Bengaluru) for helping in manuscript editing.

## Author contributions

**Conceptualization:** Sananda Biswas, Patrick D'Silva.

**Data curation:** Sananda Biswas, Patrick D'Silva.

**Formal analysis:** Sananda Biswas, Patrick D'Silva.

**Funding acquisition:** Patrick D'Silva.

**Investigation:** Sananda Biswas.

**Methodology:** Sananda Biswas.

**Project administration:** Patrick D'Silva.

**Resources:** Patrick D'Silva.

**Supervision:** Patrick D'Silva.

**Validation:** Sananda Biswas.

**Visualization:** Sananda Biswas.

**Writing – original draft:** Sananda Biswas.

**Writing – review & editing:** Sananda Biswas.

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
