## [Decision Letter · Decision Letter 0]

16 Sep 2024

Dear Dr D'Silva,

Thank you very much for submitting your Research Article entitled 'Ubp2 modulates DJ-1-mediated redox-dependent mitochondrial dynamics in Saccharomyces cerevisiae' to PLOS Genetics.

The manuscript was fully evaluated at the editorial level and by independent peer reviewers. The reviewers appreciated the attention to an important problem, but raised some substantial concerns about the current manuscript. Indeed, they unanimously recommend substantial revisions to address the concerns about data quality, statistical analysis, mechanistic insights, and overinterpretation of results. Based on the reviews, we will not be able to accept this version of the manuscript, but we would be willing to review a much-revised version. We cannot, of course, promise publication at that time.

If you decide to revise the manuscript for further consideration at PLOS Genetics, please aim to resubmit within the next 60 days, unless it will take extra time to address the concerns of the reviewers, in which case we would appreciate an expected resubmission date by email to plosgenetics@plos.org.

If present, accompanying reviewer attachments are included with this email; please notify the journal office if any appear to be missing. They will also be available for download from the link below. You can use this link to log into the system when you are ready to submit a revised version, having first consulted our Submission Checklist .

PLOS has incorporated Similarity Check , powered by iThenticate, into its journal-wide submission system in order to screen submitted content for originality before publication. Each PLOS journal undertakes screening on a proportion of submitted articles. You will be contacted if needed following the screening process.

To resubmit, log into your Editorial Manager account and select the option 'Revise Submission' in the 'Submissions Needing Revision' folder.

We are sorry that we cannot be more positive about your manuscript at this stage. Please do not hesitate to contact us if you have any concerns or questions.

Yours sincerely,

Alessia Buscaino

Academic Editor

PLOS Genetics

Eva Stukenbrock

Section Editor

PLOS Genetics

Reviewer's Responses to Questions

**Comments to the Authors:**

Reviewer #1: This is a potentially interesting manuscript that characterizes several mitochondria-related phenotypes in mutants lacking yeast DJ-1 proteins. The authors have found a genetic link between Hsp31/34 and the Ubp2 cysteine-dependent deubiquitinase. They show that loss of UBP2 restores some previously described phenotypes in strains lacking Hsp31/34. Whilst the authors have shown that loss of Ubp2 suppresses a number of phenotypes in hsp31, 34 mutants, it is not clear that this represents a physiologically significant crosstalk in wild-type cells. For example, loss of Ubp2 has similar effects in both wild-type and Δ31Δ34 strains for a number of the phenotypes examined so the authors may simply be reporting on what loss of Ubp2 causes rather than specificity for Δ31Δ34 strains. It is interesting though, since loss of Ubp2 clearly rescues the respiratory growth phenotype of Δ31Δ34 strains.

The manuscript presents a complicated data set, which at present are largely correlative. As outlined below, some conclusions are somewhat over-stated in the Discussion based on the current findings and more data are required to be so definitive. Other experiments require further analysis and controls as outlined below.

Fig. 1A – images are poor quality, particularly for SC glycerol growth at 37C. Given that the same data is shown in Fig. S1 and it is much clearer, it should be replaced in the main manuscript figure.

Figure 1B shows that Ubp2-HA is increased in Δ31Δ34 strains whereas there does not appear to be any difference in Ubp2-Ha for the mid-log phase cells shown in Fig. S2. It is not stated what growth phase was used for Fig. 1B but presumably it should have been mid-log phase to be consistent with the spot tests. This should be clarified, especially what is meant by mid-log phase (6h) and late-log phase (12h) which should be defined on growth curves. The Discussion states that DJ-1 paralogs regulate the growth phase dependent expression of Ubp2 but much more careful experiments are required to reach such a strong a conclusion. This is important since it is the only direct evidence that Hsp31, 34 affects Ubp2.

Figure 2D – Given the differential effects of Ubp2 overexpression in wild-type and Δ31Δ34 strains it is important to confirm that Ubp2 is expressed to similar levels in both strains. This control should be straight-forward using the HA tag.

The mitochondrial integrity assay shown in Fig. 2A needs more explanation. Does intermediate mean a mixture of fragmented and tubular mitochondria within cells? It would help to provide some illustrative diagrams to differentiate what is meant by fragmented, intermediate and tubular. It would be better to present the data as a percentage, presumably from triplicate experiments although this is not stated. The authors should explain how they scored the different morphologies plus discuss any potential physiological significance.

The mitophagy assay shown in Fig. 3 needs more explanation. What do the various smaller OM45-GFP fragments correspond to? Why are there 3 different size fragments. The description of this data in the text is quite subjective and it would benefit from quantification and an indication of reproducibility.

Fig. 3B legend – what do the dashed boxes and arrows indicate

The quantifications shown in Fig. 4D does not appear to match the western blots shown in Fig. 4C. For example, a very strong Fzo1 signal is detected in the ubp2 mutant at T = 0 and there is essentially no signal detected at 90 min. However, the quantification suggests that just 50% of Fzo1 remains at 90 min. The quantification also lacks statistical analysis. For this blot and others, it is not stated whether signals were normalized to loading controls as part of the quantitative analysis.

Following on from these experiments the Discussion states that the increased expression of Ubp2 contributes to the increased stability of Fzo1 in Δ31Δ34 cells – In order to reach this strong conclusion, the authors need to demonstrate that altering Ubp2 levels in a wild-type strain using their overexpression construct affects Fzo1 stability and other read-outs of mitochondrial function, dependent on the presence of Hsp31/34.

Fig. 5 and line 332 The authors suggest there is a cell cycle arrest at the G2/M phase in the Δ31Δ34 mutant. Presumably they mean there is a delay in the cell cycle at this stage. To aid a more meaningful comparison, this data would benefit from quantitation perhaps by calculating the length of time each strain spends in the different stages of the cell cycle.

As discussed by the authors, it is the GSH to GSSG ratio that is the crucial indicator for alterations in redox state and glutathione metabolism. Simply measuring GSH levels using monochlorobimaine is not sufficient to comment on changes in redox state. These assays should be repeated using approaches that measure both GSH and GSSG levels preferably using fluorescent reporter probes that report on the redox state of the different cytoplasmic and mitochondrial glutathione pools, rather than in whole cell extracts.

Reviewer #2: The manuscript “Ubp2 modulates DJ-1-mediated redox-dependent mitochondrial dynamics in Saccharomyces cerevisiae“ by Sananda Biswas and Patrick D’Silva describes a functional link between the deubiquitinase Ubp2 and DJ-1-like Hsp31 paralogues in yeast. The study provides novel insights on how Ubp2 and Hsp31 paralogues regulate mitochondrial fission/ fusion processes and give evidence for involvement of ROS production and mitophagy. Even though the manuscript delivers novel insights, some fundamental issues need to be addressed as part of a major revision. In general the statisics need to be improved and clarified for each figure. The authors tend to over-interprete the data based on single images without robust statistical evidence. The authors should only make claims where appropriate (see list). The major and minor points to be addressed are listed as follows.

Major points:

1) The authors need to give information on the statistical tests performed for each figure individually. The general statistics statement is not sufficient. Some statistical tests are missing and need to be conducted.

2) Figure 3A: The authors need to quantify the ratios of free GFP to OM34-GFP + free GFP (at least triplicates) and show additional replicates as supplemental data. The authors should be careful with their claims. It seems like the expression level of OM45-GFP is reduced in ∆ubp2.

Why are there additional bands in ∆ubp2 (below 20 kDa)?

3) Lines 273-277: It is not adequate to make such claims based on single images without any quantitative assessment. At least for ∆31∆34 and ∆31∆34 ∆ubp2 a co-staining for vacuoles should be performed to properly identify vacuolar GFP signals.

Minor points:

Data availability statement indicates that data will be accessible, but no precise platform or link is given. This needs to be completed.

Line 169: I do not observe growth impairment of ∆ubp2 at 24°C on dextrose media. Please rephrase accordingly.

Line 169-174: No hsp single mutants are shown in Fig. 1A. The reference should be made to Fig. S1.

Line 170: ∆31 show only a very weak decrease in growth.

Figure S1: The temperature needs to be properly indicated in columns.

Figure 1 C: Which statistical test was performed to calculate the p-value?

Figure S2: The bars indicating 6h and 12h are not properly positioned in the figure.

Line 188-190: To claim that Ubp2-HA levels are significantly changed in wt requires quantitative western blotting with suitable statistics.

Figure 2: It is required to mentioned the statistical tests (ANOVA?) which were performed. How was the classification of mitochondria made?

Line 226: The abbreviation NAO needs to be introduced here.

Line 228 and 231: The reference should be Fig. 2C.

Line 236/239: The reference should be Fig. 2D.

Line 237: A citation is missing.

Line 245: A space is missing between “defects(Fig. 2E)”

Lines 263-265: “∆31∆34 showed slower mitophagy induction after 54 h of growth in glycerol due to the presence of hyperfused mitochondrial structures” The authors do not show evidence for the causal link and should rephrase their claim accordingly.

Line 275: “faster” instead of “fater”

Figure 4: What kind of statistical test was performed?

Statistical analysis of 4D is missing (2-way ANOVA)

Line 330: Cite Fig. 5

Line 340: “Microscopic analysis revealed that ∆31∆34 has 30% of the cells exhibiting increased cell size than WT (more than 5-6 µm)“ There is no figure supporting this statement. The authors should show the quantification as supplemental data.

Line 347: “lelevls” to “levels”

Line 353: Introduce abbreviation for H2DCFDA.

Figure 6: Add note for statistics.

Line 358: These findings were further…

Line 369: “… the microscopic analysis was performed …

Line 376: Full stop is missing.

Line 406: Omit “with”

Line 411: To avoid confusion the authors should write “perturbation of Fzo1” instead of “perturbation of the fusion protein”

Line 433: Full stop is missing.

Reviewer #3: also as an attachment, called "Review PGENETICS-D-24-00709"

This manuscript addresses the relationship between the PARK7 paralogues Hsp31/34 and the USP10 paralogue Ubp2. In a previous study, the authors nicely showed that Hsp31 and Hsp34 are required for redox balance, mitochondrial respiration, mitochondrial morphology and cell-cycle progression. Simultaneous loss of Hsp31and Hsp34 causes a reduction in GSH levels and consequently oxidative stress, increased mitochondrial biogenesis, and mitochondrial mass, G2/M cell cycle arrest, and increased apoptosis, for the most part reverted by addition of the antioxidant NAC. Here, the authors convincingly show that ablation of the deubiquitinase Ubp2 suppresses all defects associated to loss of Hsp31/34. They also observe a correlation between Ubp2/Hsp31/34 and alterations in mitochondrial morphology and in the MFN1/MFN2 paralogue Fzo1, and propose a role of mitochondrial fusion in this process. These findings are novel and potentially very relevant. Rescue experiments and mechanistic aspects of this rescue should be substantiated, in particular investigating causal effects, as detailed below.

Main points

1) The authors nicely discover a genetic interaction between Ubp2 and Hsp31/34. In this regard:

A) They should provide evidence that re-expression of Ubp2 rescues all �hsp31�hsp32 phenotypes presented in this manuscript.

B) Moreover, is this rescue dependent on the catalytic activity of Ubp2?

C) If yes, how do the authors explain that UBP2 deletion rescues �hsp31�hsp32 phenotypes, given that Ubp2 is expected to have reduced activity in �hsp31�hsp32 cells. This prediction is because �hsp31�hsp32 cells have increased ROS levels (PMID: 37548361and 32070881) and oxidative stress inactivates the catalytic activity of Ubp2 (PMID: 25622294).

D) Along this line, repressed activity of Ubp2 is expected to cause mitochondrial fragmentation, but the �hsp31�hsp32 cells have increased mitochondrial tubulation. How do the authors explain this?

2) Related to point 1), can it be that catabolic repression caused by growth in glucose is absent in �hsp31�hsp32 cells? Consistently, mitochondrial morphology of the �hsp31�hsp32 cells intriguingly resembles WT cells in post-diauxic shift phase (PDS). Therefore, the authors should test if PDS cells resemble �hsp31�hsp32 cells in cell cycle progression, mitophagy, ROS levels or growth resistance to H2O2. If yes, are these effects in PDS cells rescued by deletion of UBP2?

3) Related to point 2) is the rescue of respiration, mitochondrial mass, cell cycle arrest and ROS levels observed in �ubp2�hsp31�hsp32 cells rather due to effects in the TCA and ETC pathways? The authors could follow an approach similar to https://doi.org/10.1101/2024.07.09.602707. For example, is the rescue dependent on the carbon source? What happens upon deletion of RTG1 or Hap4 overexpression?

4) �hsp31�hsp32 cells have altered levels and stability of Fzo1. How do these chaperones affect Fzo1? The authors should test if this is caused by an increase in mitochondrial biogenesis.

Of note, in PMID: 32070881 they observe that �hsp31�hsp32 cells have increased transcriptional of all 3 major mt dynamics proteins: Dnm1, Mgm1 and Fzo1, and very comparable increases in their respective protein levels. Consistently, in this same manuscript, they show that �hsp31�hsp32 cells have increased mitochondrial mass and increased GSSG levels, thus altering the GSH/GSSG ratio. Reduction in GSH is an indicator of oxidative stress and they could revert increased mitochondrial mass with the antioxidant NAC.

5) They infer that UBP2 deletion rescues the cellular defects of �hsp31�hsp32 cells due to the ubiquitination, stability and steady state levels of Fzo1. However, besides presenting correlative data, they do not demonstrate any causal effect. Therefore, at this point, the many statements along the manuscript inferring causality are misleading (for example title of Fig. 4, statement in lines 345-346, 379-380, 410-411, 436-437, 476, 480-482, 485).

To investigate if rescue depends on mt dynamics, the authors should simultaneous eliminate Fzo1 and Dnm1, which allows to maintain mitochondrial tubulation and respiration. They should compare respiration, mitochondrial mass, cell cycle arrest and ROS levels of��ubp2�hsp31�hsp32 cells and of �fzo1�dnm1�ubp2�hsp31�hsp32 cells.

Of note: �ubp2 cells have fragmented mitochondria, but do not affect mitochondrial mass, cell cycle progression, ROS levels or growth resistance to H2O2. This suggests that mitochondrial alterations per se do not cause cell cycle, ROS or mitochondrial mass defects.

Other points:

6) To my knowledge, a causal effect of mitochondrial fusion and fission in cell cycle progression has not been demonstrated in S. cerevisiae, which compromises a few statements, e.g. lines 320-321. Please comment/correct.

5) In their redox biol manuscript (PMID: 32070881), the authors show that �hsp31�hsp32 cells have growth defects on glycerol similar to �hsp31�hsp34. However, only the last one has increased mitochondrial mass and post-diauxic-shift-like mitochondria, while �hsp31�hsp32 cells have wild-type like mitochondria. Therefore, the different phenotypes related to Hsp31 loss are not necessarily related. Please comment in the manuscript.

6) In their mitophagy assessments, it would be advisable to perform FM464 co staining with OM45-GFP to demonstrate interiorization in to the vacuole.

7) It could be preferable to name WT-morphology as tubular, and �hsp31�hsp34 as PDS-like.

8) The introduction could be shortened to the most relevant functions of Hsp31 and Ubp2 in context of this manuscript, given that no clinical aspects are addressed in the present study.

**Have all data underlying the figures and results presented in the manuscript been provided?**

Reviewer #1: **No: ** No primary data is included

Reviewer #2: **No: ** The original data is not provided.

Reviewer #3: Yes

PLOS authors have the option to publish the peer review history of their article (what does this mean? ). If published, this will include your full peer review and any attached files.

**Do you want your identity to be public for this peer review?** For information about this choice, including consent withdrawal, please see our Privacy Policy .

Reviewer #1: No

Reviewer #2: No

Reviewer #3: No

---

## [Decision Letter · Decision Letter 1]

29 Jan 2025

PGENETICS-D-24-00709R1

Ubp2 modulates DJ-1-mediated redox-dependent mitochondrial dynamics in Saccharomyces cerevisiae

PLOS Genetics

Dear Dr. D'Silva,

Thank you for submitting your manuscript to PLOS Genetics. As with all papers, your manuscript was reviewed by members of the editorial board. After careful consideration of the revised manuscript and the reviewers' comments, we regret to inform you that we have decided to **reject** the manuscript. While two of the reviewers provided overall positive feedback, the concerns raised by Reviewer #3 remain substantial, and we do not believe the current version of the manuscript adequately addresses these critical points.

Reviewer #3 highlighted several essential areas requiring further work, particularly the lack of convincing **rescue experiments** , the need for **causal validation of key findings** , and inconsistencies in the **interpretation of results** . Furthermore, many of the requested revisions were either not performed or not sufficiently addressed,  This revised version was **not ready for publication** , and we must emphasize that revisions are meant to **improve manuscripts and make them suitable for publication.**

We encourage you to submit a **new version of the manuscript in the future** once the additional and necessary experimental work has been completed and the concerns raised by the reviewers have been comprehensively addressed. We are sorry that we cannot be more positive on this occasion. We very much appreciate your wish to present your work in one of PLOS's Open Access publications. Thank you for your support, and we hope that you will consider PLOS Genetics for other submissions in the future.

Yours sincerely,

Alessia Buscaino

Academic Editor

PLOS Genetics

Eva Stukenbrock

Section Editor

PLOS Genetics

Aimée Dudley

Editor-in-Chief

PLOS Genetics

Anne Goriely

Editor-in-Chief

PLOS Genetics

**Additional Editor Comments (if provided):**

**Reviewers' Comments (if peer reviewed):**

Reviewer's Responses to Questions

**Comments to the Authors:**

Reviewer #1: I am satisfied that the authors have made the requested revisions to their manuscript apart from the following minor corrections:

Comment 2 - The author’s response has clarified that Ubp2 expression is altered at the late log phase, as presented in Fig.1B and Fig.S1. However, this has not been clarified in the manuscript itself or legend for Figure 1 to state what growth phase was used. It is important to clarify this since the increase in Ubp2 expression is not observed at the same growth phase as that used for the spot tests

Comment 5 - The authors have added quantification for the mitophagy assay shown in Fig. 3 as a supplementary figure. This should be added as a panel to main Figure 3 along with appropriate statistics.

Comment 8 – The authors have now included data describing the GSH:GSSG ration in mitochondria and cytosol. The Methods would benefit from more description of how these experiments were actually performed, particularly how mitochondrial and cytosolic extracts were prepared and verified

Reviewer #2: The authors have addressed all of the raised criticism in a point by point reply and have presented an improved manuscript. Minor issues which need adaptation from my point of view are limited to Fig. 7, where some pictures seem inapproprately contorted/ stretched and some labelling behind white boxes remains visible and Suppl. Fig. 3 where the white boxes do not fit to the 2x-zoom pictures. Also here it seems that some of the pictures are slightly contorted/stretched.

Reviewer #3: in the attachment

**Have all data underlying the figures and results presented in the manuscript been provided?**

Reviewer #1: Yes

Reviewer #2: Yes

Reviewer #3: None

PLOS authors have the option to publish the peer review history of their article (what does this mean? ). If published, this will include your full peer review and any attached files.

**Do you want your identity to be public for this peer review?** For information about this choice, including consent withdrawal, please see our Privacy Policy .

Reviewer #1: No

Reviewer #2: No

Reviewer #3: No

---

## [Decision Letter · Decision Letter 2]

5 Jun 2025

PGENETICS-D-24-00709R2

Ubp2 modulates DJ-1-mediated redox-dependent mitochondrial dynamics in Saccharomyces cerevisiae

PLOS Genetics

Dear Dr. D'Silva,

Thank you for submitting your manuscript to PLOS Genetics. After careful consideration, we feel that it has merit but does not fully meet PLOS Genetics's publication criteria as it currently stands. Therefore, we invite you to submit a revised version of the manuscript that addresses the points raised during the review process.

Please submit your revised manuscript within 30 days Jul 05 2025 11:59PM. If you will need more time than this to complete your revisions, please reply to this message or contact the journal office at plosgenetics@plos.org. Please include the following items when submitting your revised manuscript:

We look forward to receiving your revised manuscript.

Kind regards,

Alessia Buscaino

Academic Editor

PLOS Genetics

Eva Stukenbrock

Section Editor

PLOS Genetics

Aimée Dudley

Editor-in-Chief

PLOS Genetics

Anne Goriely

Editor-in-Chief

PLOS Genetics

**Reviewers' comments:**

Reviewer's Responses to Questions

**Comments to the Authors:**

Reviewer #1: I am satisfied that the authors have made the requested revisions to their manuscript

Reviewer #2: My points have been addressed in the previous round of review, which is why my criticism is limited to the new sections:

The new figure 3B is missing a Y-axis title. What is shown here? Is it the ratio of free GFP to Om45-GFP + free GFP? Is this normalised to the loading control? This also needs to be described in the methods section. The description of the antibodies (name, no., company) which were used is missing as well. Writing "the respective primary and secondary antibodies" is not enough.

In the figure legend of figure 3 the last mentioning of (B) has to be changed into (C).

The authors should make sure that all the adaptations from peer review are properly and correctly processed. Special care is required here.

**Have all data underlying the figures and results presented in the manuscript been provided?**

Reviewer #1: Yes

Reviewer #2: Yes

PLOS authors have the option to publish the peer review history of their article (what does this mean? ). If published, this will include your full peer review and any attached files.

**Do you want your identity to be public for this peer review?** For information about this choice, including consent withdrawal, please see our Privacy Policy .

Reviewer #1: No

Reviewer #2: No

**Figure resubmission:**
---

## [Editor Report · Decision Letter 3]

12 Jun 2025

Dear Dr D'Silva,

We are pleased to inform you that your manuscript entitled "Ubp2 modulates DJ-1-mediated redox-dependent mitochondrial dynamics in Saccharomyces cerevisiae" has been editorially accepted for publication in PLOS Genetics. Congratulations!

Yours sincerely,

Alessia Buscaino

Academic Editor

PLOS Genetics

Geraldine Butler

Section Editor

PLOS Genetics

Aimée Dudley

Editor-in-Chief

PLOS Genetics

Anne Goriely

Editor-in-Chief

PLOS Genetics

Comments from the reviewers (if applicable):

**Data Deposition**

http://datadryad.org/submit?journalID=pgenetics&manu=PGENETICS-D-24-00709R3

**Press Queries**

---

## [Editor Report · Acceptance letter]

PGENETICS-D-24-00709R3

Ubp2 modulates DJ-1-mediated redox-dependent mitochondrial dynamics in Saccharomyces cerevisiae

Dear Dr D'Silva,

We are pleased to inform you that your manuscript entitled "Ubp2 modulates DJ-1-mediated redox-dependent mitochondrial dynamics in Saccharomyces cerevisiae" has been formally accepted for publication in PLOS Genetics! Your manuscript is now with our production department and you will be notified of the publication date in due course.

With kind regards,

Lilla Horvath

PLOS Genetics

On behalf of:
